# A Novel Microwave Hot Pressing Machine for Production of Fixed Oils from Different Biopolymeric Structured Tissues

**DOI:** 10.3390/polym15102254

**Published:** 2023-05-10

**Authors:** Sherif S. Hindi, Uthman M. Dawoud, Iqbal M. Ismail, Khalid A. Asiry, Omer H. Ibrahim, Mohammed A. Al-Harthi, Zohair M. Mirdad, Ahmad I. Al-Qubaie, Mohamed H. Shiboob, Najeeb M. Almasoudi, Rakan A. Alanazi

**Affiliations:** 1Department of Agriculture, Faculty of Environmental Sciences, King Abdullaziz University (KAU), Jeddah 21589, Saudi Arabia; 2Department of Chemical and Materials Engineering, Faculty of Engineering, King Abdullaziz University (KAU), Jeddah 21589, Saudi Arabia; 3Department of Chemistry, Faculty of Science, Center of Excellence in Environmental Studies, King Abdullaziz University (KAU), P.O. Box 80216, Jeddah 21589, Saudi Arabia; 4Department of Environment, Faculty of Environmental Sciences, King Abdullaziz University (KAU), Jeddah 21589, Saudi Arabia

**Keywords:** fixed oil, hot pressing machine, microwave irradiation, methylation, GC/MS

## Abstract

A microwave hot pressing machine (MHPM) was used to heat the colander to produce fixed oils from each of castor, sunflower, rapeseed, and moringa seed and compared them to those obtained using an ordinary electric hot pressing machine (EHPM). The physical properties, namely the moisture content of seed (MCs), the seed content of fixed oil (S_cfo_), the yield of the main fixed oil (Y_mfo_), the yield of recovered fixed oil (Y_rfo_), extraction loss (EL), six Efficiency of fixed oil extraction (E_foe_), specific gravity (SG_fo_), refractive index (RI) as well as chemical properties, namely iodine number (IN), saponification value (SV), acid value (AV), and the yield of fatty acid (Y_fa_) of the four oils extracted by the MHPM and EHPM were determined. Chemical constituents of the resultant oil were identified using GC/MS after saponification and methylation processes. The Ymfo and SV obtained using the MHPM were higher than those for the EHPM for all four fixed oils studied. On the other hand, each of the SG_fo_, RI, IN, AV, and pH of the fixed oils did not alter statistically due to changing the heating tool from electric band heaters into a microwave beam. The qualities of the four fixed oils extracted by the MHPM were very encouraging as a pivot of the industrial fixed oil projects compared to the EHPM. The prominent fatty acid of the castor fixed oil was found to be ricinoleic acid, making up 76.41% and 71.99% contents of oils extracted using the MHPM and EHPM, respectively. In addition, the oleic acid was the prominent fatty acid in each of the fixed oils of sunflower, rapeseed, and moringa species, and its yield by using the MHPM was higher than that for the EHPM. The role of microwave irradiation in facilitating fixed oil extrusion from the biopolymeric structured organelles (lipid bodies) was protruded. Since it was confirmed by the present study that using microwave irradiation is simple, facile, more eco-friendly, cost-effective, retains parent quality of oils, and allows for the warming of bigger machines and spaces, we think it will make an industrial revolution in oil extraction field.

## 1. Introduction

Oil-producing crops have been attracting more attention due to their high profit compared to other crops. Many species are suitable to be a source of fixed oils, for example, but not limited to castor bean, sunflower, rapeseed, moringa, oil palm, macauba palm, babassu palm, buriti palm, pequi, oiticia, coconut, avocado, Brazil nut, macadamia nut, jatrova, jojoba, pecan, bacuri, ghoper plant, pissava, olive tree, opium poppy, peanut, cocoa, linseed, sesame, and tung oil tree [1].

Fixed oils are synthesized in plants in special biopolymeric tissues of the oily plants and accumulate in their seeds. These seeds differ anatomically (histologically) from each other [2,3,4,5,6]. Different studies have been undertaken to know the morphological characteristics of oily seeds, showing that the structure (macro or micro) of raw materials is a key parameter to understanding the effect of processing and the final yield and quality of products [3,5].

Lipids are generally stored as triacylglycerols in oil bodies. Although there are quantitative differences in the accumulation of storage reserves in seeds, it was not clear whether this would also qualitatively affect the fatty acid profiles of triacylglycerols in these seeds [5].

It was reported by Hu et al. [5] that lipids and protein are major storage reserves in mature Brassica seeds. An inverse relationship between oil and protein accumulation in seeds has been reported for some plant species, including rapeseed. In addition, seeds of many plant species, including rapeseed, accumulate lipids to supply the energy requirements for germination and seedling growth. Such lipids are generally stored as triacylglycerols in small, spherical, and discrete intracellular organelles called oil bodies. Accordingly, the greater the number of oil bodies, the higher the oil content of a seed [5]. In *moringa oleifera*, lipid bodies surrounded the protein bodies and filled most of the remaining space in all healthy cells [6].

The castor bean (*Ricinus communis* L.), also called the castor oil plant, is an annual flowering plant in the Ricinus genus. There are large populations of this plant throughout tropical and subtropical regions, and it has also been found in agricultural sites in Saudi Arabia [7]. Due to its unique ability to withstand thermal fluctuations, castor oil is a valuable raw material in the pharmaceutical, medical, food, chemicals, agricultural, textile, paper, plastic, rubber, cosmetics, perfumeries, electronics, paints, inks, additives, lubricants, and biofuel industries [8,9,10,11,12,13]. According to [14,15,16], castor oil has different chemical compositions and physical–chemical properties according to its origin. Malaysian castor seeds, for instance, yield castor oil with a saponification value of 182.96 mg KOH/g as well as a 43.3% dry weight [14], while Nigerian castor seeds yield castor oil with a saponification value of 178.00 mg KOH/g and a dry weight of 48%. In addition, its viscosity and specific gravity are high, making it soluble in alcohols with very low solubility in petroleum solvents [14,15,16,17]. Seed oil content ranges from 30% to 60% by weight with variations according to variety, geographic location, and extraction method [11,18,19]. The castor oil’s high boiling and low melting points make it superior to petroleum-based oils particularly at high and low temperatures [17,18,19].

Castor oil consists primarily of ricinoleic acid which has three functional groups. These functional groups are responsible for regulating the chemical composition of the oil which is used to produce resins, waxes, polymers, and plasticizers. It has been demonstrated that carboxylic groups can give rise to a variety of esterification products, while the point of unsaturation can be modified by hydrogenation, epoxidation, or vulcanization [20,21]. Additionally, this oil is important because of its hydroxyl groups and double bonds in the ricinoleic acid component [16]. As a result of the presence of the hydroxyl group in RA and its derivatives, the oil is oxidatively stable with a number of facilitated chemical reactions, including halogenation, dehydration, alkoxylation, esterification, and sulfation [9,22]. In addition to comprising esters (triglycerides molecules) comprised of 3-carbon alcohols (glycerol) and 3-carbon fatty acids (18 carbon atoms), such as other vegetable oils [11,15,23], castor oil is also unique among vegetable oils due to the presence of the hydroxylated fatty acid ricinoleic acid which is the only source of this fatty acid commercially available [11]. It is well known that castor oil contains a wide range of beneficial compounds for health, including fatty acids, flavonoids, phenolic compounds, amino acids, terpenoids, and phytosterols [24]. There are a number of important fatty acids detected in castor oil, including linoleic, oleic, stearic, and linolenic fatty acids [11,22]. It can be used as a bio-based lubricant obtained from the fatty acids of castor oil [25].

The sunflower (*Helianthus annuus* L.) belongs to the family Asteraceae and originated in eastern North America. It is grown throughout the world as one of the most widely cultivated oil crops due to its relatively short growing season [1] and its ability to grow in large semi-arid regions without irrigation [26,27]. It has been found that the oleic acid content of commercial hybrids of sunflower varies between 10 and 50% depending on the climatic conditions of the field and the temperature at which the seeds are growing. Several studies have demonstrated a strong negative correlation between linoleic acid and oleic acid [1,28]. On average, this variety contains 75% oleic acid, although individual plants range between 50 and 80% [29] and individual seeds vary between 19 and 94% [30]. There were significant increases in oleic content in offspring of this variety even when temperatures varied with values exceeding 83% under various conditions [30,31].

Rapeseed is an essential crop for many different purposes, including edible oil, biodiesel, lubricant, and feed. The oil of this plant is high in oleic acid which gives it a competitive advantage over other cooking oils [32]. In terms of oil content, rapeseeds varied between 30.6% and 48.3% of their dry weight which was found to be a higher percentage compared with canola seed oils. The most abundant fatty acids are oleic (56.80–64.92%) and linoleic (17.11–20.92%) as indicated by Matthaus et al. [33] with an omega-6 to omega-3 ratio of 2:1 which offers cardiovascular health benefits [34]. High levels of tocopherols, polyphenols, phytosterols, and carotenoids have been reported in cold-pressed rapeseed oil. Due to its rich content of antioxidants and nutraceuticals, this oil possesses anti-hypercholesterolemic and anti-inflammatory properties. As a result of the high nutritional value of cold-pressed rapeseed oil, it can be used in foods, cosmetics, and pharmaceuticals [34].

*Moringa oleifera* Lam. is a widely distributed species of the Moringaceae family and is a fast growing, hardwood tree native to the sub-Himalayan area of Northern India [35,36]. In addition to its nutritional and medicinal value, the plant contains phytochemicals that are effective antioxidants and antimicrobials [37]. In addition to their high oil content (up to 40%), seed kernels also contain high levels of fatty acids (oleic acid > 70%) as well as exceptional resistance to oxidative damage. In fact, this rich oil profile makes moringa seeds ideal for both human consumption and commercial use [38,39]. Seeds, on the other hand, have attracted scientific attention due to their high fatty acid content and after refining, significant resistance to oxidative degradation [39]. About 76.73 percent of the oil consists of monounsaturated fatty acids, mainly oleic acid, as well as gadoleic acid and palmitoleic acid. It contains 21.18% saturated fatty acids of which palmitic acid is the predominant followed by behenic, stearic, and arachidic acids. In addition, small quantities of cerotic, lignoceric, myristic, margaric, and caprylic acids have been reported. As a whole, the content of polyunsaturated fatty acids is very low, on average 1.18%, while that of linoleic and linolenic acids is 0.76% and 0.46%, respectively. Moreover, the extraction method does not appear to significantly impact the oil’s fatty acid composition. Only one study reported a modest increase in myristic acid and stearic acid content in solvent-extracted oil compared to cold-pressed oil [40].

There are two methods for extracting fixed oils from seeds: mechanical pressing (cold or hot), solvent extraction, or a combination of the two [41]. The mechanical pressing method has the disadvantage of extracting less oil from seeds (about 45%), making it necessary to re-extract the remaining oil from the seed cake. On the other hand, the high-temperature hydraulic pressing method was found to be effective in extracting about 80% of the oil [42]. Moreover, cold-pressed castor oils have low acid and iodine values and have a slightly higher saponification value than solvent-extracted oils [11] obtained a maximum yield of 50.16% of castor oil by pressing for 60 min at 108 °C, using isopropanol/hexane (50:50 *v*/*v*) as a solvent system.

Using electric heating tools, such as coils or bands require thick cords, complicated circuits, consumes more electric energy, and can only heat the space covered by the coil or band. Contrarily, microwave offers a thermally homogeneous spaces in any desired size [43,44,45,46,47,48,49]. In addition, the ordinary available electric hot presser machines depend for their heat transferring, mainly on the conduction phenomenon that increases the oil extraction durations and reduces the extraction efficiency itself.

This study was initiated to evaluate the yield and properties of each of the four fixed oils squeezed using two machines that differed in their heating tools (EHPM and MHPM) to ensure that the novel microwave procedure used did not distort the parent quality of fixed oils.

## 2. Materials and Methods

### 2.1. Management Plan

As shown in Appendix A presented in the supplementary material, testing the efficiency of the novel microwave hot pressing machine in extracting high yield and quality of fixed oils was applied according to the following steps:

Selection of the species.

1.Specifying the seeds amounts required for the extraction processes.2.Preparation of the electric hot pressing machine by inserting the electric heating coil around the extruder head of the machine.3.Preparation of the microwave hot pressing machine by releasing the electric heating coil from the extruder head of the machine.4.Pressing the seeds of each of the four oil crops (castor, sunflower, rapeseed, or moringa).5.Dehydration, settlement, and filtration of the crude fixed oils.6.Characterization of the fixed oils and comparing the oil extracted by microwave to that obtained by electric band heater.

### 2.2. Raw Materials 

Four different species were used to be the source of fixed oils, namely castor bean (*Ricinus communis* L., Family: Euphorbiaceae), common sunflower (*Helianthus annuus* L., Family: Asteraceae), rapeseed (*Brassica napus* L., Family: Brassicaceae), and moringa (*Moringa oleifera* Lam., Family: Moringaceae) as shown in Appendix A. A random sample of three shrubs/plants of each species was collected during the 2021–2022 season from the Agricultural Research Center of King Abdulaziz University at Hada Al-Sham which is located about 120 km from Jeddah, Saudi Arabia. The shrubs/plants were grown in sandy soil at a latitude of 21° 46′.839N and a longitude of 39° 39′.911E which was 206 m above sea level. 

#### 2.2.1. Preparation of the Seeds

Appendix A illustrates the steps involved in the extraction of each of the four fixed oils from the air-dried seeds of castor bean (Appendix A), common sunflower (Appendix A), rapeseed (Appendix A), and moringa (Appendix A). A bean de-huller machine (Appendix A) was used to separate the seeds from their hulls/pods after being collected from the selected shrubs/plants, and the seeds were, then, cleaned to remove any foreign materials [7].

#### 2.2.2. The Mechanical Extraction of the Four Fixed Oils

For each castor bean, common sunflower, rapeseed, and moringa, about one kg of the clean seeds were air dried to reduce the moisture of the seeds [48,50,51]. The air-dry seeds were allowed to be oven dried at about 65–70 °C in an electric furnace. To reduce the oil viscosity within seeds tissues that can maximize the oil yield [52,53], the seeds were exposed to a microwave beam for about ten minutes within the seed hopper. Then, the seeds were hot-pressed mechanically using each of the two hot pressing machines (MHPM and EHPM) as shown in Figure 1. After settling for one hour, the oily supernatants were collected individually and filtered through vacuum filters to remove contaminants from the oil. Hydraulic pressure was used to release the remaining oil from the settled cake [50].

#### 2.2.3. The Hot Pressing Machine (HPM)

Microwave radiation is generated using the magnetron device, the essential component of the microwave generator unit (MGU), by converting the input alternate electric current (AC) into microwave beam that is responsible for warming the pressing machine. The HPM (Figure 1) consists of a motor, extruder, colander, and a heating tool (electric heater or microwave beam). It is worth mentioning that two devices were used in the present investigation for squeezing each of the four oily seeds, namely electric hot pressing machine (EHPM) and microwave hot pressing machine (MHPM). Both of them use the same hot pressing machine but they differ in the heating tool necessary to warm the extruder’s colander of the machine. For more illustration, the EHPM uses a dc-electric band heater (Figure 2a–c) for this purpose, while the MHPM was constructed to be dependent on a microwave generator unit that emitters microwave beams responsible to release the colander temperature. 

The main components of the HPM are the Taby motor (380 Volt 3 phase 10 amp 50/60 Hz, Sweden) that rotates the extruder shaft (Figure 2a,b) of the HPM. The extruder shaft is used to deliver the oily seeds from the seed hopper to the colander, in which the seeds are compressed mechanically, allowing the fixed oil to excrete from biopolymeric tissues of seeds through the oil channels of the colander as clear from Figure 1 and Figure 2. The power screw geometry of its straight shaft is apparent in Figure 2b. Heating the oily seed tissue found within its inner central channel allowing for extraction of their fixed oil and excreted through its oil capillary channels where they were collected in a suitable reservoir. In addition, the colander is the terminal part of the pressing machine coupled directly to the extruder (Figure 1a,b, Figure 2a,c, Figure 3a,c and Figure 4e). It has a hollow central tunnel (to feed the extracted seed tissues known as the cake) as well as an oil channel that accumulates the final oil yield in the production reservoir. To soften seeds tissues and enhance fixed oils yield, a heating source must be used to worm the colander of the HPM. In this investigation, two heating techniques were performed and compared to each other, namely electric band heater used for EHPM and MGU used for MHPM.

To prepare the EHPM, the electric heating coil must be installed, while for installing the microwave hot pressing, the electric band heater must be released from its colander and replaced with the MGU in which the microwave beam was directed to the extruder’s colander by using an ideal waveguide, a short metallic waveguide (Figure 1b and Figure 4e). 

The band heater (Figure 3b,c) was used to heat the colander that is coupled to the extruder front. It is also suitable for heating any metal pipes, nozzles, barrels, and other cylindrical parts. These fast-heat band heaters electrically warm the external surface of a cylindrical part to provide indirect heating. It is worth mentioning that the heating process is an essential facility to prevent plugging the fine holes that exert the oil, reducing the oil viscosity for ease of handling within the extruder as well as increasing the oil yield released from the seed tissues. 

In the MGU, the magnetron is the primary and only source of microwave energy (Figure 4a,d). Originally, it was taken from a domestic microwave oven. In terms of technical specifications, the magnetron has a power of 900 W, an anode voltage of 4.20 kVp, and a frequency of 2.46 GHz. The magnetron, 2M214 39F (06B), code of 2B71732E for LG microwave ovens (900 W, 4.20 kVp anode voltage, 2460 MHz frequency) was used in this study as a self-excited microwave oscillator, cavity magnetrons convert high-voltage electric energy into microwave beams (Figure 4a,d). It is worth mentioning that magnetrons produce high-power outputs by crossing electron and magnetic fields. In this study, a magnetron emits a constant frequency of 2.46 GHz, since magnetrons are usually designed on a constructively fixed frequency as illustrated by Hindi et al. [45,46,47,48].

As shown in Figure 1b and Figure 4e, microwave energy is radiated into the air through a stainless-steel pipe with a diameter of 1 inch, then is conducted into a metal cavity. In order to melt the fixed oil and force it to excrete from seed tissues, temperature must be adjusted to be approximately 75–80 °C.

The high voltage transformer (HVT) is used to convert alternate current (AC) into direct current (DC) required for working the electric band heater. In addition, the high voltage transformer (Teknik, Amal, Sweden) was used in the present investigation as shown in Figure 4b,d. It has been reported that large industrial and commercial ovens can use 915 megahertz magnetrons to excite the larger cavities within the ovens [49].

The high voltage capacitor (2100V.AC, 1%, 50/60 Hz, and 10 MΩ). An electrical connection was made between the capacitor and the magnetron, the transformer, as well as the outlet of the waveguide via a diode (Figure 4c).

Other accessories were used in the MGU, such as an electronic diode, the double-terminal component in which the current conducts primarily in one direction (asymmetric conductance). Furthermore, its resistance is low in one direction (ideally zero) and high in the other direction (ideally infinite). In addition, a small electric fan was constructed to distribute air around the transformer and magnetron to cool them. 

### 2.3. Characterization of the Four Species

After collecting the crude oil, it was weighed and stored until it was subjected to different characterizations. A set of oil specifications were determined using methods reported in the standard methods or applied by researchers working in the same or related fields [50,51,52,53,54]. 

The values of the different physical and chemical properties of the seeds and fixed oils of the four crops species were calculated as presented in Table 1. 

#### 2.3.1. Physical Characterization of the Seeds and Oils

The physical properties of fixed oils are known to be influenced by chemical composition as well as operating parameters, such as pressure and temperature. These properties were determined and compared with the cited standard [7,51].

Concerning seed qualifications related to their suitability for producing fixed oils, moisture content (MC) as well as seed content of fixed oil (S_cfo_) were determined. For the MC investigation, about 100 g of cleaned intact seeds were dried in an oven at 90 °C, weighed at 1 h intervals until obtaining a constant weight after about 6 h [42]. It is worth mentioning that the S_cfo_ refers to the amount of oil that can be derived from an oil seed. İt is usually represented as a percentage based on the parent seed weight as shown in Table 1 [52].

Regarding fixed oil extraction parameters, four properties were determined, namely yield of the main fixed oil (Y_mfo_), yield of recovered fixed oil (Y_rfo_), extraction loss (EL), and efficiency of fixed oil extraction (E*_foe_*). 

The Y_mfo_ refers to the quantity of oil that can be extracted from an oil seed. İt was determined as the ratio of the weight of main oil extracted to the weight of the ground seed specimen before extraction. Furthermore, it was determined mechanically using each of the MHPM and EHPM, while both Y_rfo_ and EL were investigated by solvent extraction procedures using analytical grade n-hexane anhydrous (95% purity) purchased from Sigma-Aldrich, Germany in a Soxhlet extractor.

Prior to the chemical extraction performed, only for the purpose of estimating both Y_rfo_ and EL, the air dry seeds were oven dried at about 65–70 °C in a heating furnace. After that, the oven-dried seeds were ground using a rotary grinding machine and sieved using standard sieves to be passed through a standard sieve of 60 mesh and retained on that of 80 mesh (60/80 mesh). This range of the seeds’ particle size facilitates the diffusion of the soluble compounds and the release of the oil [52,53].

The suitable particle-sized seed powder was put in a cotton thimble (2.5 × 7 cm) and centered in one siphon of the Soxhlet extractor fixed with a round flask (250 mL) containing about 150 mL of n-hexane. After that, a rotary evaporator with a vacuum pump was also used to separate the oil and the solvents after oil extraction. The recovered fixed oil was calculated in the manner indicated in Table 1.

Extraction loss (EL) was calculated as the difference between the weight of the ground seeds specimen before extraction and the sum overall weights of both the oil recovered and residual cake after extraction by hexane divided by the parent weight of the ground seed sample before extraction as clear in Table 1 [52].

The E*_foe_*, an important parameter in oilseed processing, is the percentage of oil extracted in relation to the amount of oil present in the seed. It was computed as the ratio of the weight of oil recovered to the product of the seed oil content and the weight of the crushed seed sample before extraction [52]. The E*_foe_* was computed as a ratio of the weight of oil recovered to the product of the seed oil content and the weight of the crushed seed sample before extraction [52]. 

Regarding the specific gravity of fixed oil (SG_fo_), in a clean 25 mL oven dried, stoppered bottle, the fixed oil was pyrchonometrically determined at 25 °C. The weight of the bottle was measured before and after it was filled with the fixed oil. Following the washing and oven drying of the bottle, the fixed oil was substituted with deionized water and the bottle was weighed (W_2_). The SG_fo_ is calculated based on the equation shown in Table 1 [7,51,55,56].

Moreover, the refractive Index (RI) provides a rapid indicator of its purity and quality. A refractometer model No. 922313 (Bellingham and Stanley Ltd., London, UK) was used to determine the RF of the castor oil at 30 °C. For each fixed oil, three repetitions were performed and recorded [51,57,58].

#### 2.3.2. Chemical Characterization of the Fixed Oils

##### Determination of Iodine Number (IN)

An IN represents the amount of unsaturationality (the presence of double bonds) in a fixed oil determined by the amount of iodine that reacts with 100 g of oil. Oils with a higher IN contain more double bonds [59]. The iodine value is counted as g I_2_/100 g of oil.

The IN was analyzed using the method described by AL-Hamdany and Jihad [60] and Omari et al. [51]. After dissolving the fixed oil in 10 mL of chloroform, the solution was mixed with 30 mL of Hanus iodine solution, kept in a tightly closed flask, and vigorously swirled for 30 min in the dark. Thereafter, approximately 10 mL of potassium iodide (15%) and 100 mL of distilled water were added under shaking. Titration was carried out with the iodine solution against sodium thiosulfate (0.1N) until a yellowish color was achieved. Approximately two to three drops of the starch solution were then added, turning the solution color blue. Titration continued until the blue color faded, and the volume of Na_2_S_2_O_3_ at this point was calculated. A blank test (without oil) was conducted in the same manner. In order to calculate the iodine number (IN), the formula presented in Table 1 was used [7,51,53].

##### Determination of Saponification Value (SV) 

By measuring the amount of potassium hydroxide required to saponify one gram of fixed oil, SV provides an indication of the chain length of the fatty acids contained in the oil. Based on the standard procedure described by Akpan et al. [57], the SV was determined. About 2 g of a fixed oil was added to 25 mL of 0.1 N ethanolic-KOH with constant stirring was allowed to be boiled gently for 60 min under refluxing. A few drops of phenolphthalein indicator was added to the warm mixture and then titrated with HCl (0.5 M) up to the end point in which the pink color of the indicator just disappeared. Based on the equation presented in Table 1, we were able to determine the saponification value. 

##### Determination of Acid Value (AV) 

The AV of an oil is defined as the milligram amount of potassium hydroxide required to neutralize one gram of the oil. The term is used to indicate how many carboxylic acid groups (-COOH) a fatty acid contains. In order to determine the AV of fixed oil, 0.5 g of the oil was dissolved in 25 mL of absolute ethanol phenolphthalein indicator. A shaking water bath was used to heat the mixture for five minutes. Afterward, a drop-wise titration with KOH (0.1 N) was conducted while the free acid was still hot. The solution was continuously agitated until a pink color appeared which remained for a minimum of 10 s. Upon approaching the endpoint, vigorous shaking was performed to ensure thorough mixing. During the reaction with the oil, KOH (0.1N) was measured in terms of volume based on the formula shown in Table 1 [51,53].

##### Determination of pH Value

In a beaker with a slow stirring, about 2 g of each fixed oil was mixed with 13 mL of hot distilled water. Afterward, the emulsion was cooled in a cold-water bath until it reached 25 °C. Using suitable buffer solutions, an electrode was standardized and immersed in oil to determine its pH value [51,55].

##### Determination of the Fatty Acids of the Fixed Oils

Two samples were analyzed for each crop, whereby one of them was extracted by the electric hot pressing machine (EHPM), and the other was produced by the microwave–hot pressing machine (MHPM). Statistically, three repetitions were performed to detect the experimental error.

To ensure high-accuracy chemical analyses, the fixed oil samples were pretreated as the following scheme before characterization by using the GC/MS (Appendix A).

1.Preparation of the Methyl Esters

Using silica gel columns, each fixed oil was eluted with hexane [7,61,62]. Under reduced pressure, hexane was evaporated, resulting in the so-called hexane fraction (Appendix A).

The procedure includes the following steps: saponification, esterification, fractionation on a column chromatography filled with silica gel, and analysis by GC/MS. To remove the free lipids, about 2 g of fixed oil was extracted with 10 mL mixture of CHCl_3_/MeOH solution (2:1 *v*/*v*) and sonicated for about 15 min, each. To release “bound” components, the free lipids-fixed oil sample was saponified with 0.5 M NaOH (MeOH/H_2_O solution, 9:1 *v*/*v*, 5 mL, 70 °C, 1 h).

After centrifugation of the cold mixture at 2500 rpm for 15 min, the supernatant was discarded, the precipitate was collected, and its pH was readjusted to 3 by adding 3 M HCl. About 1 mL of solvent-extracted water and 3 mL of CHCl_3_ were used to extract the hydrolyzed lipids. After the complete elimination of the solvent by evaporation under a steady flowing helium atmosphere, the hydrolyzed acid components were methylated using 100 L of BF_3_/MeOH (14% *w*/*v*) at 70 °C for 1 h). Then, the methyl esters were extracted with about 2 mL of CHCl_3_ (3 times) from the solvent-extracted H_2_O.

Using a glass column filled with activated silica gel preheated at 120 °C for 24 h, the methylated extracts were fractionated and eluted using n-hexane. 

The extracts were separated using the elutropic series of the following four solvent systems, including n-hexane, dichloromethane (DCM), and methanol (MeOH). The four systems were: 6 mL of n-hexane, 2 mL of a mixture of n-hexane/dichloromethane (9:1 *v*/*v*), 6 mL of dichloromethane, and 5 mL DCM/MeOH (1:1 *v*/*v*) to yield an elution rate of 15 mL min1. The DCM/MeOH method allowed the dihydroxy-fatty acids to be eluted. Under a mild steady stream of helium gas, all fractions were collected, and the solvent was removed.

2.Gas Chromatography-Mass spectrometer (GC–MS)

Since saponification/methylation processes (Appendix A) are helpful for identifying fatty acids [62,63], the major constituents of the oil were detected as methyl or methyl esters derivatives. GC-Trace ultra-system (Thermo Co. USA) was used to analyze each of the studied fixed oils. Fractionation and separation were conducted using a fused-silica capillary column (Elite- 5MS column, Thermo Scientific GC Column, 30 m × 0.25 mm × 0.25 μm) with chemically bonded phases DB1 (J&W Scientific). The initial temperature (IT) was adjusted to 40 °C for 4 min as residence time (RT) and raised to a maximum final temperature (MFT) of 220 °C at a rate of 4 °C/min for 15 min. The temperature was set at 250 °C for the injector, where the injection volume was 1 μL of the tested sample at a transfer temperature of 280 °C. Helium was used as a carrier gas at a flow rate of 20 mL/min. The EI mass spectra were collected in full scan mode (*m/z* 50–600) at 180 °C ion source temperature and 70 eV ionization voltage. Based on a library search using NIST, USA, and Wiley Registry 8-edition as well as a comparison of retention indices among the peaks, tentative identifications were made. A peak area normalization method was to estimate the yield of fatty acids (Y_fa_) in the fixed oil [7,64] by calculating the peak area (in mm^2^) of a certain fatty acid detected at a certain retention time (in minutes) based on the overall areas in mm^2^ representing chemical constituents of the fixed oil that arisen at different retention times (in minutes) at the chromatogram as clear in Table 1. Moreover, their chromatograms are presented in Appendix A. The percentage allocations of the fatty acids in the fixed oils are presented in Figure 5.

### 2.4. Scanning Electron Microscopy (SEM)

SEM imaging was used to study the surface morphology and anatomical features in biopolymeric structured tissues. The SEM technique and image analysis were efficient tools for the characterization of the microstructure of the four oily seeds [3,4,5].

Sample preparation was conducted by isolating a small central block of cotyledon (about 3 mm in length) from an air-dried seed.

For the SEM examination, the sample was placed on a double side carbon tape on Al-stub and dried in air. It is worth mentioning that all samples were examined without any pretreatments, such as embedding, fixation and sputtering processes [3]. The samples were examined with an SEM Quanta FEG 450, FEI, Amsterdam, The Netherlands. The microscope was tried at an accelerating voltage ranging from 5–20 kV. 

### 2.5. Statistical Design and Analysis

The present experiment was statistically designed to be a split plot design in three replicates (blocks) with multiple observations (five repetitions) per sub-plot per block for each property of each fixed oil. This was conducted to detect the differences between the four fixed oils produced from castor, sunflower, rapeseed, and moringa seeds squeezed by the hot pressing machine heated with two different heating tools, namely electric band heater (EHPM) and microwave irradiation (MHPM). In addition, the least significant difference at a 95% level of confidence (LSD_0_._05_) method was used to compare the differences between species means for all the properties studied [65].

## 3. Results

The physical and chemical properties of each of the four fixed oils were determined. The comparisons between the mean values of these properties were extended to those within species as well as between species. Accordingly, the comparisons were conducted between the fixed oils extracted using the microwave–hot pressing machine (MHPM) and the electric hot pressing machine (EHPM) as well as between the oily species at the same level of machinery used.

### 3.1. Characterization of the Fixed Oils

Eleven physical properties of the four fixed oils, namely the moisture content of the seed, ^2^ seed content of the fixed oil, ^3^ yield of the main fixed oil, ^4^ yield of recovered fixed oil, ^5^ extraction loss, ^6^ efficiency of fixed oil extraction, ^7^ specific gravity, ^8^ iodine number, ^9^ saponification value, ^10^ acid value, ^11^ and yield of fatty acid were determined and are presented at Table 2. Furthermore, the chemical properties of these oils, namely iodine number (IN), saponification value (SV), acid value (AV), and pH were also investigated and included in Table 2. 

#### 3.1.1. Physical Properties of Seeds

Investigation of the moisture content of seeds (MC_s_) revealed that the castor beans had the lowest MC_s_ (3.6%), while moringa had the highest MC_s_ value (5.34%) among the four oily species as is clear in Table 2. The obtained mean values approach those determined by Muzenda et al. [42] and differed from those reported in the literature findings of 4.15% [66,67,68]. Moreover, comparing the two heating techniques (MHPM and EHPM) revealed that the MHPM dried all four seeds to lower levels compared to the EHPM.

For the seed content of the fixed oil (S_cfo_), comparing the S_cfo_ values between species indicate that the castor bean seeds contained the highest S_cfo_ value (49.87%) than the other species. Moreover, the MHPM extracted more fixed oils than those obtained by the EHPM (Table 2). 

#### 3.1.2. Physical Properties of the Fixed Oils

For the yield of the main fixed oil (Y_mfo_), it is clear from Table 2 that at the same level of heating technique (MHPM and EHPM), comparing the Y_mfo_ values within species revealed that castor seeds gave the highest yield (44.8 % and 38.3 %, respectively) comparing to those obtained from sunflower (42.5 % and 36.4 %, respectively), rapeseed (42.2 % and 35.2 %, respectively), and moringa (35.6 % and 34.6 %, respectively) as presented in Table 2. At the same level of oily species, the MHPM produces higher Y_mfo_ values than those obtained by the EHPM (Table 2). This is clear for all the four species examined in the present study. 

Regarding the yield of the recovered fixed oil (Y_rfo_), it can be seen from Table 2 that the Y_rfo_ obtained by the MHPM (6.93–8.85%) that was higher than those for the EHPM (3.34–5.06%).

Studying the extraction loss (EL) indicates that the results of MHPM (1.02–2.03%) were lower than those obtained from the EHPM (2.35–2.75%). This finding confirms the superiority of the invented MHPM. In addition, comparing species at the same level of the heating tool revealed that there are no significant differences between them.

For the efficiency of the fixed oil extraction (E_foe_), the MHPM had the highest efficiency (90.22–92.8%) compared to those for the EMHPM. Moreover, comparing species at the same level of the hot pressing procedure revealed that there are no significant differences between them.

##### Specific Gravity of Fixed Oil (SG_fo_)

For the SG_fo_ property, there are statistical differences between the SG_fo_ values among the four species examined, and the comparisons made within or between species. For more illustration, at the same level of heating tool (MHPM and EHPM), comparing the SG_fo_ values within species revealed that castor seeds gave the highest yield (0.973 and 0.968, respectively) comparing to those obtained from sunflower (0.914–0.918, respectively), rapeseed (0.906 and 0.914, respectively), and moringa (0.946 and 0.957, respectively) as presented in Table 1. The mean SG_fo_ of castor oil was found to be 0.973. However, the mean value of castor oil SG_fo_ determined represents a slight increase over the ASTM specification range (0.957–0.968).

At the same level of oily species, the MHPM produces nearly identical values for those obtained by the EHPM (Table 2). This indicates that microwave irradiation did not alter the SG_fo_ of the resultant fixed oils protruding the economic importance of this novel investigation.

The refractive index (RI) of the fixed oil was also examined in this study. It was found that there are no statistical differences between the RI values among the four species examined as well as the comparisons made within each of the four oily species. At the same level of the hot pressing tool (MHPM and EHPM), statistical comparisons of the RI values within species revealed that castor seeds gave nearly the same index value (1.48 and 1.43, respectively) compared to those obtained from sunflower (1.473 and 1.469, respectively), rapeseed (1.465 and 1.467, respectively), and moringa (1.464 and 1.563, respectively) as presented in Table 2. Moreover, at the same level of oily species, the MHPM produces nearly identical values to those obtained by the EHPM (Table 1). The RI for castor oil (1.48), is within the range of values cited [51]. This indicates that microwave beams did not cause any difference in the RI of the resultant fixed oils. This result confirms the suitability of the present study for the present industrial application.

#### 3.1.3. Chemical Properties of the Fixed Oils

Concerning the iodine number (IN), it was found that at the same level of pressing machines (MHPM), sunflower (126.3) and rapeseed (114.5) seeds gave higher INs than those for castor oil (83.7) and moringa (69.7). Furthermore, for the IN values obtained using EHPM, the same trend was repeated with an identical sequence as follows: Sunflower (127.1) and rapeseed (115.1) seeds gave higher INs than those for castor oil (84.8) and moringa (71.4). In addition, within species, there are no significant differences between using MHPM and EHPM among all the species studied (Table 2).

As shown in Table 2, although the IN values for castor oil (83.7 and 84.8, respectively) were lower than 100 IN-unit for both MHPM and EHPM; these values are located at the ASTM-specification limit. On the other hand, those values for moringa fixed oil for MHPM and EHPM (69.7 and 71.4, respectively) were lower than the ASTM specification range. 

For the saponification value (SV) of the fixed oil, examining Table 2 revealed that sunflower oil had the highest SV value for both MHPM and EHPM (188.4 and 194.6, respectively) compared to the other resources examined, namely castor bean (181.5 and 182.1, respectively), rapeseed (168.3 and 181.2, respectively), and moringa (180.7 and 190.4). In addition, microwave beams produced a slight reduction in the SV compared to the action of the electric band heater.

Moreover, speculating the acid value (AV) revealed that at the same level of the heating techniques (MHPM and EHPM) comparing the AV values within species revealed that castor seeds gave the highest AV (44.8% and 38.3%, respectively) compared to those obtained from sunflower (42.5% and 36.4%, respectively), rapeseed (42.2% and 35.2%, respectively), and moringa (35.6% and 34.6%, respectively) as presented in Table 2.

At the same level of oily species, the MHPM produces higher AV values than those obtained by the EHPM (Table 2). This is clear for all the four species examined in the present study. 

In comparison to the literature [42,51], the AV of castor oil (0.869 mg KOH/g oil) lies within ASTM-specification guidelines (0.4–4.0 mg KOH/g oil). 

The pH value of the fixed was achieved to estimate the oil’s dissociation state since fixed oils may contain free fatty acids. As shown in Table 2, there are no significant differences between the pH values, and for both comparisons made within or between species. The pH values lie within the normal range although it is slightly higher than those found by Omari et al. [51], especially for castor oil.

Through the GC/MS analysis of the fixed oil, the main fatty acids constituting castor bean, sunflower, rapeseed, and moringa oils examined in this study were identified (Figure 5 and Appendix A). The mass spectra and fragmentation patterns of the predominant fatty acids constituents were determined by comparison with the mass spectra stored in different available databases [69].

Examining Figure 5 concerning the four fixed oils resulting from the microwave–hot pressing machine (MHPM) and the electric hot pressing machines (EHPM) revealed that four fatty acids were detected in all four fixed oils but in different concentrations.

1.For castor oil (Figure 5a_1_,a_2_), ricinoleic acid was the most abundant fatty acid (76.41 and 71.99%) among the four fixed oils resulting from MHPM (Figure 5a_1_) and EHPM (Figure 5a_2_), respectively. However, castor oil had the lowest fatty acid content due to lower contributions from oleic acid (3.31%), linoleic acid (4.32%), and palmitic acid (4.32%).

It is worth noting that the high concentration of ricinoleic acid (76.41%) in the present study corresponds with the range (70–90%) cited by numerous researchers [11,22,70].

The amount of oleic acid in MHPM and EHPM was approximately 5.2% and 4.02%, respectively (Figure 5a_1_,a_2_). In addition, linoleic acid accounted for approximately 4.38% and 3.49% of the total fatty acids in the MHPM and the EHPM, respectively (Figure 5a_1_,a_2_). Furthermore, the amount of palmitic acid in the castor oil was approximately 7.16% and 5.08%, respectively. There may be an economic value to its sodium salt which is used in the soap and cosmetic industries [58,71].

2.For the sunflower fixed oil (Figure 5b_1_,b_2_), the oleic acid was the prominent fatty acid with high contents of 67.5% and 61.1% for the MHPM and EHPM, respectively. In addition, linoleic and palmitic acid had lower contents in the sunflower oil.3.For the rapeseed fixed oil (Figure 5c_1_,c_2_), oleic acid, linoleic acid, and palmitic acid were the major fatty acids detected in the rapeseed fixed oil (65.7%, 20.1%, and 6.32%, respectively) produced by the MHPM, while their yields were 62.1%, 18.08%, and 4.08%, respectively, by using the EHPM.4.For the moringa fixed oil (Figure 5d_1_,d_2_), oleic acid and palmitic acid were found in the moringa fixed oil in concentrations of 76.86% and 5.36%, respectively, using the MHPM, while their contents were 72.49 and 4.08% for the EHPM.

Based on the data presented in Figure 2, it can be concluded that the prominent fatty acid of the castor oil is ricinoleic acid, making up 76.41% and 71.99% of the MHPM and EHPM, respectively. In addition, the oleic acid was the prominent fatty acid in each of the fixed oils of sunflower, rapeseed, and moringa species (65.7–76.86% for the MHPM and 62.1–72.49% for the EHPM).

The chemical structure of the compound along with the mass spectra-chromatogram of the four fatty acids detected at the four fixed oils extracted are shown in Appendix A patterns [7,72,73,74]. 

To confirm the chemical structures of the prominent fatty acid (ricinoleic acid) in the castor oil, its molecular mass and fragmentation pattern were studied and presented in Appendix A. Based on the mass spectrum, the prominent peaks were observed at 28, 29, 41, 43, 55, 67, 69, 74, 82, 83, 84, 87, 96, 97, 98, 124, 166, and 198 *m/z*. The presence of the compound was confirmed by comparing its mass spectrum with that of the standard 9-octadecenoic acid, 12-hydroxy-, methyl ester, and [R-(Z)]-profile (Appendix A) as both compounds exhibit similar fragmentation.

In addition, the molecular mass and fragmentation pattern was studied to confirm the chemical structure of the prominent fatty acid (oleic acid) detected in the fixed oils of sunflower, rapeseed, and moringa species and presented in Appendix A. According to the mass spectrum, prominent peaks were observed at 27, 29, 41, 43, 55, 83, and 97 *m/z*. The existence of the compound was confirmed by comparing its electron ionization chromatogram with the standard (9Z)-octadec-9-enoic profile.

Furthermore, the chemical structure of linoleic acid was confirmed using its mass spectra chromatogram (Appendix A). It was sorted as the second prominent fatty acid in the sunflower fixed oil, the prominent peaks were clearly present at 29, 41, 55, 67, 68, 81, 96, 110, and 124 *m/z*. Comparing its mass spectrum with its standard profile, the compound existence was confirmed where they have the same fragmentation patterns. All four fixed oils contain palmitic acid (Appendix A) in minor amounts. 

A mass spectrum of the sample revealed prominent peaks at 41, 43, 56, 74, 87, and 143 *m/z*. The existence of this compound was confirmed by comparing its mass spectrum with the standard hexadecanoic acid and methyl ester profile. 

### 3.2. Effect of Microwave Irradiation on Biopolymeric Structured-Tissues (BST)

Technical information relevant to the microwave irradiation device that replaced ordinary electric heating coils to warm the hot pressing machine (HPM) essential for extracting fixed oils from the BST is presented in Figure 1, Figure 2, Figure 3, Figure 4 and Figure 6. This figure shows the microwave generator unit (MGU), termed as a high-voltage magnetron, the high voltage-transformer, the high voltage-capacitor, and the manner for directing the microwave beam to the colander of the microwave hot pressing machine (MHPM) through a special waveguide.

Furthermore, the theoretical behavior of the microwave beam (Figure 4) illustrates the sinusoidal wave curve (SWC), energy level along with the SWC, and the proportionality between square amplitude and the energy carried by the wave [46,47,48].

For studying the biopolymeric structured tissues as affected by microwave irradiation, the four oily seed precursors were anatomically examined by SEM and are presented in Figure 7. These micrographs show the cellular structure of each castor bean (*Ricinus communis*, common sunflower (*Helianthus annuus*), rapeseed (*Brassica napus*), and moringa (*Moringa oleifera*). Furthermore, Figure 7 shows the endosperm cells (EC), cell walls (CW), cytoplasm (CY), compound lipid bodies (CLBs), and subcellular organelles of singular lipid bodies (SLBs).

For comparing the oily bodies inserted in the four structured tissues, their estimated results were obtained using image analysis and presented at Table 3.

## 4. Discussion

### 4.1. Physical Characterization of Physical Characterization of the Fixed Oils

1.The moisture content of seed (MC_s_) values were found to be varied in seeds that ranged from 3.6% to about 7% [42,67]. The variation could be attributed to the difference in the nature of beans from different locations [42] and/or seed macro-structure, including hull-to-kernel weight ratio, hull thickness, and their oil content [66].2.Focusing on the seed content of fixed oil (S_cfo_) showed that the recovery of the remained traces in the seeds’ cake by using solvent extraction procedure allowed the yield of recovered fixed oil (Y_rfo_) to be maximized as well as reducing the extraction loss (EL) for the fixed oils. Some remaining traces of fixed oils in the cakes can be attributed to incomplete cell lysis within the seeds which possibly trapped and retained some amount of oil [52].

The efficiency of fixed oil extraction (E_foe_) for the MHPM was higher than that for the EHPM. 

A specific gravity of fixed oil (SG_fo_) can be used as an indicator of its purity and as a tool to distinguish a variety of oily solutions as stated by Omari et al. [51]. Furthermore, they noted that the SG_fo_ of oil floats over an aqueous solution when a spill occurs where compounds are found in water as a result of the spill [51]. Due to the use of the hot pressing extraction technique in the present study, some impurities were present in the castor oil that may have contributed to the minor differences in the SG_fo_ values between literature studies [51].

It has been reported by Omari et al. [51] that minor differences between cited data in the RI are due to differences in planting and harvesting conditions of the mother shrubs as well as research conditions. A number of impurities may alter the RI values, including gums, phosphates, etc. [51]. Castor oil’s RI is generally related to its degree of unsaturation or conjugation and vice versa [51]. Due to the two direct relationships between the un-saturationality and the iodine number and refractive index, its moderate RI value for the castor oil matches its moderate iodine number (84.8 iodine unit).

Since the MHPM produces nearly identical values for those obtained by the EHPM at the same level of oily species (Table 2), it can be concluded that microwave beams did not cause any difference for the RI of the resultant fixed oils. This result confirms the suitability of the present study for the present industrial application.

### 4.2. Chemical Characterization of the Fixed Oils

Iodine number, saponification value, acid value, and yield of fatty acid of the four fixed oils were studied to ensure whether microwave irradiation may influence their chemical quality as well as fatty acids yield.

The lower IN values can be attributed to the presence of more saturated fatty acids which did not react with the Hanus iodine solution [51]. In contrast, higher IN values for sunflower and rapeseed (>100) indicate higher levels of unsaturation which increases the amount of iodine absorbed by unsaturated acids which makes them ideal as drying oils in the cosmetic, varnish, and coating industries as well as to make cosmetics.

Castor and moringa oils, however, had INs lower than 100 iodine units and could be classified as non-drying oils, incompatible with paint industries but suitable for soap industries [75].

According to Omari et al. [51], higher SV indicates lower molecular weights (MW) of triglycerides and vice versa. As a result, castor oil was expected to contain a lower MW of triglycerides than other fixed oils with a lower SV. It was shown in Table 2 that castor oil had a high SV, which was within the ASTM specification range of 175–187 SV-units and comparable to those reported in the literature [51,76,77]. It is possible to attribute the variations seen between the cited data to the small differences in the fatty acid composition of castor oil as these values are characteristic of castor oil.

The castor oil had low free fatty acid content since the AV is an indicator of the number of carboxylic acid groups in fatty acids constituting an oil and it is also a measure of the amount of free fatty acids. This study obtained a lower AV for castor oil than that of Omari et al. [51], as the seeds had been collected from the ground following curing for a period of time. Thus, the seeds had been exposed to sufficient amounts of lipase enzyme to hydrolyze their triglycerides into free fatty acids, thus improving their acid content.

Among the four fixed oils examined, the higher pH value of castor oil is expected to be attributed to its high free fatty acid content which is correlated with its acid value [51].

These fatty acids are represented as methyl or methyl esters due to pretreatments of the fixed oils using the saponification/methylation route. It has been demonstrated that methylation of oil causes the oil to become more volatile which subsequently adapts the Programme Temperature Volume of the injector of the GC-MS device [71,74]. There is no difference in fragmentation patterns between methyl esters derivatives of free fatty acids and their parent free fatty acids except that methyl esters have a higher molecular weight than their parent free fatty acids by 14 g/moles [74].

Castor oil is a unique source of ricinoleic acid, a hydroxylated fatty acid (Figure 5), it can cause the oil to be converted from a non-drying oil to a drying oil by removing the hydroxyl groups in its chain. This is useful in the manufacture of alkyd resin coating [74] and adds to its industrial advantages. 

### 4.3. Effect of Microwave Irradiation on Biopolymeric Structured-Seed Tissues

Several researchers have been trying to use microwave irradiation, a sustainable non-contact heating source, for oil extraction from aromatic and oily plants [48,78,79,80,81,82,83]. The magnetron receives energy from a high-voltage transformer (HVT). Regarding technical specifications, the HVT is 1000E-1E, 220 V and 60 Hz [48]. In order to achieve the required 2.45 GHz, the magnetron converts the high voltage alternate current (AC). It has been reported that large industrial and commercial ovens can use 915 megahertz magnetrons to excite the larger cavities within the ovens [46,48]. It is worth mentioning that wavelengths of microwaves range from approximately one meter to one millimeter with frequencies between 300 MHz (1 m) and 300 GHz (1 mm). While electromagnetic waves clearly exhibit wave characteristics, they also exhibit particle characteristics at high frequencies [48,75].

As shown in Figure 6a–c, the microwave beams can pass through oily seed tissues via the microwave generator. There is usually a frequency of 2.46 GHz (with a wavelength of 12.24 cm) for microwave beams. In microwave heating, microwaves penetrate the interior of biopolymeric structured–tissues, including but not limited to seeds as well as foods affecting their dipolar molecules of water, fats, or sugars. Accordingly, heat is generated because dipolar molecules absorb electromagnetic radiation which leads to their intensive vibration producing friction that leads to rapid temperature rising and, consequently, efficient water evaporation and/or melting fats. This results in a greatly increased vapor pressure differential between the center and surface of the biopolymeric tissue, allowing fast transfer of moisture and or essential oil out of the tissue. Hence, microwave drying as well as the microwave hot pressing technique is rapid, more uniform, and energy efficient compared to ordinary ones [48,84,85]. Through their electric and magnetic fields, electromagnetic (sinusoidal) waves (Figure 6a) are able to bring energy into a system [46,47,48]. In addition to exerting force and moving charges in the system, these fields may also perform work on the charges. It is much more efficient to transfer energy when the electromagnetic wave frequency matches the natural frequency of the system (such as microwaves at the resonant frequency of water molecules). The energy of a wave is proportional to the square root of its amplitude (Figure 6c). A larger E-field and a larger ß-field exert greater forces and are capable of performing more work with electromagnetic waves.

Furthermore, microwave beams have hot and cold spots which makes them unsuitable for heating the extruder’s colander (Figure 6b). By facing a microwave beam with damp thermal paper, this phenomenon can be detected. It is apparent from examining the propagation line (the baseline) of the microwave sinusoidal curve that cold (damping) spots are analogous to the intersection points of both the magnetic and electric wave curves. With our invention, the hot spots have been considered and eliminated since the extruder rotates continuously, alternating the damping spots.

For the EHPM, heat energy and entropy are transferred mainly by conduction according to Fourier’s law. On the other hand, for the MHPM, heat is transferred mainly via conduction (according to the Fourier’s law), convection (according to Newton’s law of cooling), and some amount of heat is transferred via radiation (Kirchhoff’s law of thermal radiation).

Conduction heat is responsible for moving heat from the surficial regions of the oily seed tissues to their cores, while convection combines conduction heat transfer and circulation to force molecules in the air to move from the hottest zones to cooler ones. Moreover, radiation is the process where heat and light waves strike and penetrate your food. As such, there is no direct contact between the heat source and the cooking food.

Accordingly, the microwave hot pressing machine is an ideal tool for heat transfer and, subsequently, heating the oily seed tissues increases the fixed oil yield [45,84].

### 4.4. Effect of Ultrastructure of Seeds on Fixed Oils Extraction

Understanding the microstructure of different oily seed organelles would be helpful to isolate and characterize their lipids more efficiently. In particular, the knowledge of the seed microstructure may be important in industrial processing [3], especially for detecting the suitability of exchanging microwave irradiation with convenient heating techniques in the oil extraction process.

Histological features of the four oil species are shown in Figure 5 for castor bean (*Ricinus communis* L.), (b) Common sunflower (*Helianthus annuus* L.), (c) Rapeseed (*Brassica napus* L.), and (d) Moringa (*Moringa oleifera* Lam.).

It is clear from Figure 7 that there are two spherical-shaped biopolymeric organelles spreading in the endoplasm termed novelly as compound LB (CLB) and singular LB (SLB) as clear in Figure 7A–D. The difference between the CLB and SLB is that the former contains huge internal spheres of LB, while the SLB is a single individual oily body.

The difference between the four species was estimated by analyzing their images for CLB (Figure 7a_1_–d_1_) and SLB (Figure 7a_2_–d_2_).

For the castor bean, the CLB’s diameter was found to be ranged from 6.67 to 22.2 µm, while the SLB’s diameter varied from 0.44 to 0.92 µm. Furthermore, regarding the sunflower biopolymeric tissue, CLB’s diameter ranges from 8.7 µm to 22.09 µm and that for SLB differed from 0.19 µm to 0.47 µm. In addition, examining the diameters of the lipid organelles of the rapeseed revealed the CLB’s diameter ranged from 7.27 µm to 14.55 µm, while the SLB’s diameter differed between 0.97 µm to 0.59 µm. For the lipid body found immersed in the endosperm of the moringa seed, it was found that the diameter of the CLB (6.52 µm–17.39 µm) was higher than that for the SLB (0.24 µm to 0.71 µm) as shown in Table 3 and Figure 7.

Lipids are generally stored as triacylglycerols in oil bodies. Although seeds differed quantitatively in the accumulation of storage reserves, it was not clear whether this would also qualitatively affect the fatty acid profiles of triacylglycerols in these seeds [5].

In addition, examining Figure 7a_1_–d_1_, the cells are configured in a rectangular shape which can be better packaged than spherical cells in the endosperm [3]. These results were from the approach found by other researchers [3,4,5,6].

It can be seen from Table 3 and Figure 7a_1_–d_1_ that castor bean has the highest SLB diameter (0.44 µm–0.94 µm) followed by those moringa seeds. This finding is agreed with that found and reported by 3. Perea-Flores et al. [3] for the castor oil crop as well as other species [3,5,6,7]. Although our presentations (Table 3 and Figure 7) covered the differences between the four examined biopolymeric structured tissues [7,48,51,74,86], they did not give impressions concerning the overall fixed oil content of them. This explains the highest fixed oil yield obtained from such species.

Compared with conventional methods, microwave treatment for oil extraction has many advantages, e.g., improvement of extracted oil yield and quality, direct extraction capability, lower energy consumption, faster processing time, and reduced solvent contents [7,80]. These results may be attributed to microwave irradiation which provides a potential alternative to induce stress reactions in ultrastructured tissues within the oil seeds. By using microwave radiation in oil seeds, a higher extraction yield and an increase in mass transfer coefficients can be obtained because the cell membrane is more severely ruptured. Apart from this, permanent pores were generated accordingly, this enables the oil to move through permeable cell walls [42,44].

## 5. Conclusions

1.A microwave beam was used to heat the extruder’s colander of a hot pressing machine instead of the ordinary electric one.2.The invented microwave–hot pressing machine was used to produce, individually, four different fixed oils extracted from seeds of castor, sunflower, rapeseed, and moringa species and compared to those obtained using the ordinary electric–hot pressing machine.3.The physical properties, namely moisture content, oil content, oil yield, oil extraction efficiency, specific gravity, and refractive index, were determined for the four fixed oils.4.The chemical properties, namely iodine number, saponification value, acid value, pH, and chemical constituents, using gas chromatography coupled with a mass spectrometer of the four oils extracted by using both heating tools were evaluated based on those in the literature for the four fixed oils.5.The higher oil extraction efficiency indicated that using microwave irradiation enhanced the oil yield with retaining the parent quality of the fixed oils that is very encouraging for candidating such an invention as a pivot of the industrial fixed oil projects.6.Studying the histological features of the biopolymeric structured tissues revealed that castor bean species has the highest singular lipid body diameter (0.44 µm–0.94 µm) followed by those for moringa seeds. This explains the highest fixed oil yield obtained from such species.7.Comparing with conventional methods, microwave treatment for oil extraction has many advantages, e.g., improvement of extracted oil yield and quality, direct extraction capability, lower energy consumption, faster processing time, and reduced solvent contents [7,75].

## Figures and Tables

**Figure 1 polymers-15-02254-f001:**
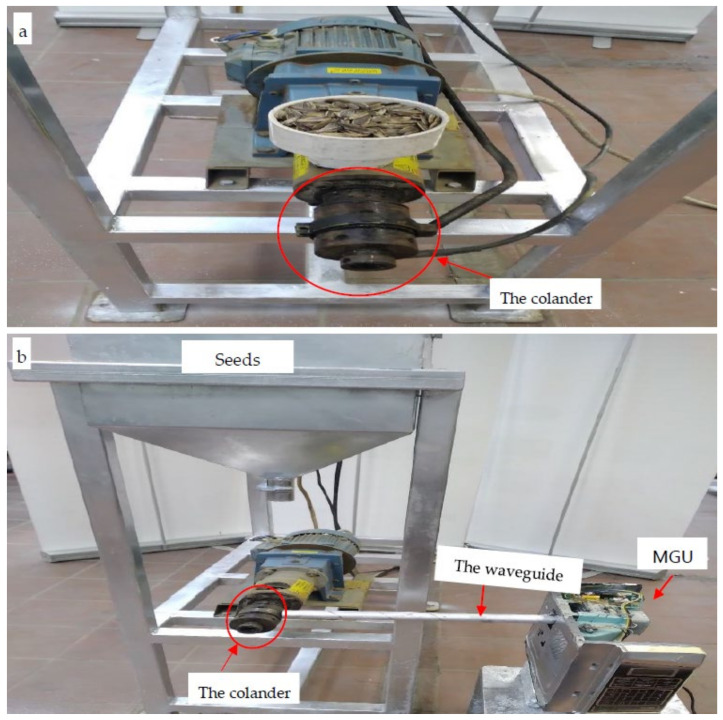
The microwave hot pressing machine (MHPM) used for extraction of the four fixed oils: (**a**) An overall front image for the HPM before connecting the microwave source, (**b**) An overall front image for the MHPM showing the colander, the waveguide that used for directing microwave beam to the extrusion head, and the microwave generator unit (MGU).

**Figure 2 polymers-15-02254-f002:**
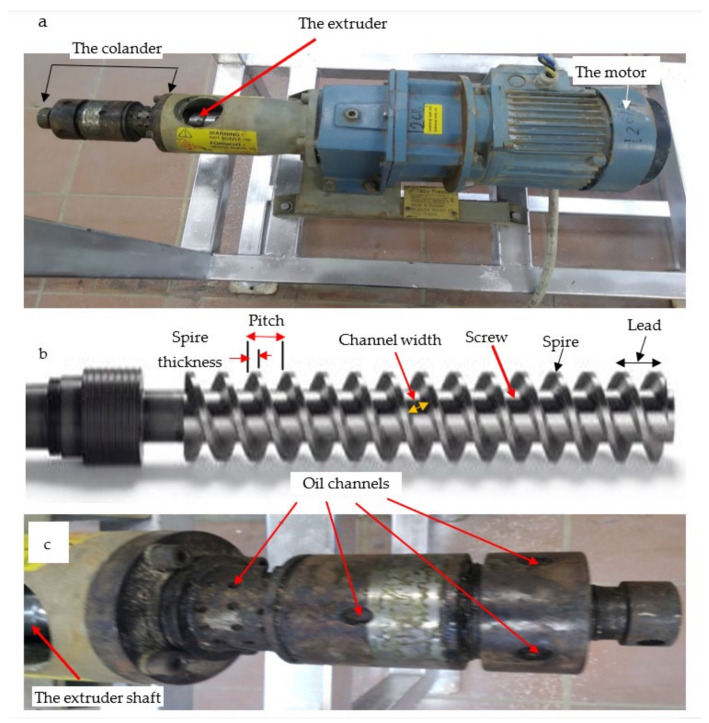
The hot pressing machine (HPM): (**a**) Upper view, (**b**) The straight shaft and its power screw geometry, and (**c**) The colander of the hot pressing machine.

**Figure 3 polymers-15-02254-f003:**
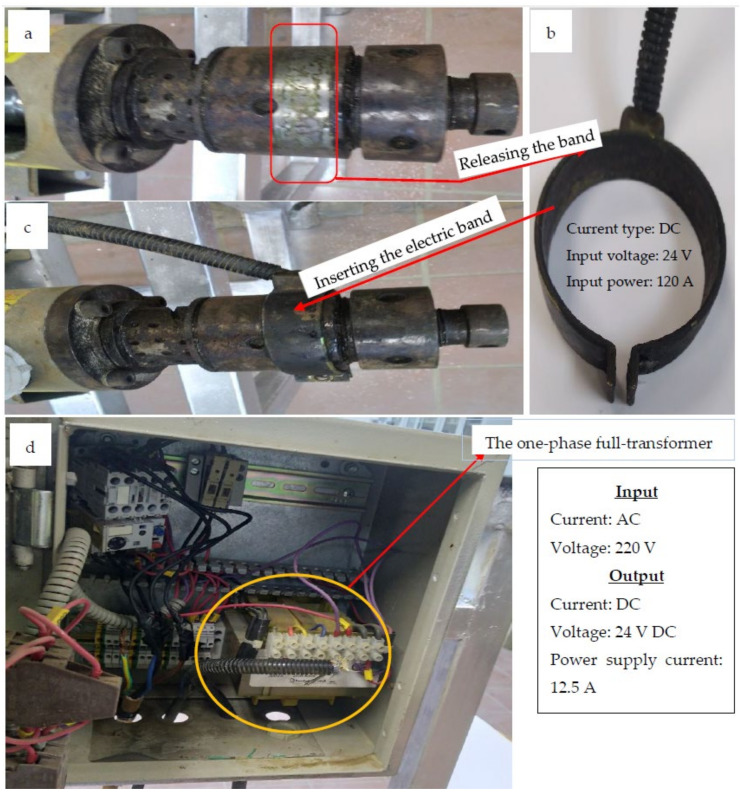
Removing the dc-electric band from the EHPM to be exchanged to the MHPM: (**a**) The DC-electric band heater used to heat the colander of the extruder, (**b**) The thermal band, (**c**) Fixing the band tighten around the colander, and (**d**) The electric box feeding the HPM.

**Figure 4 polymers-15-02254-f004:**
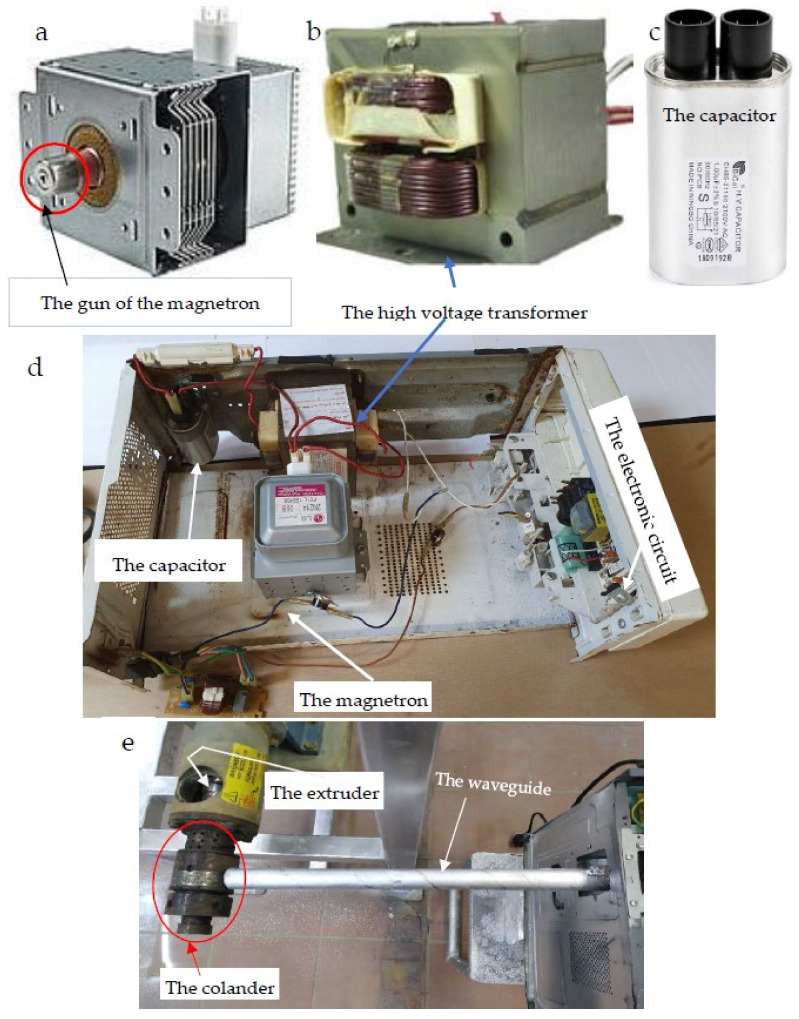
The microwave generator unit (MGU) used for heating the head of the microwave hot pressing machine: (**a–e**): (**a**) The high-voltage magnetron, (**b**) The high-voltage transformer, (**c**) The high-voltage capacitor, (**d**) Overall image of the MGU, and (**e**) Directing microwave beam to the colander of the MHPM through the waveguide.

**Figure 5 polymers-15-02254-f005:**
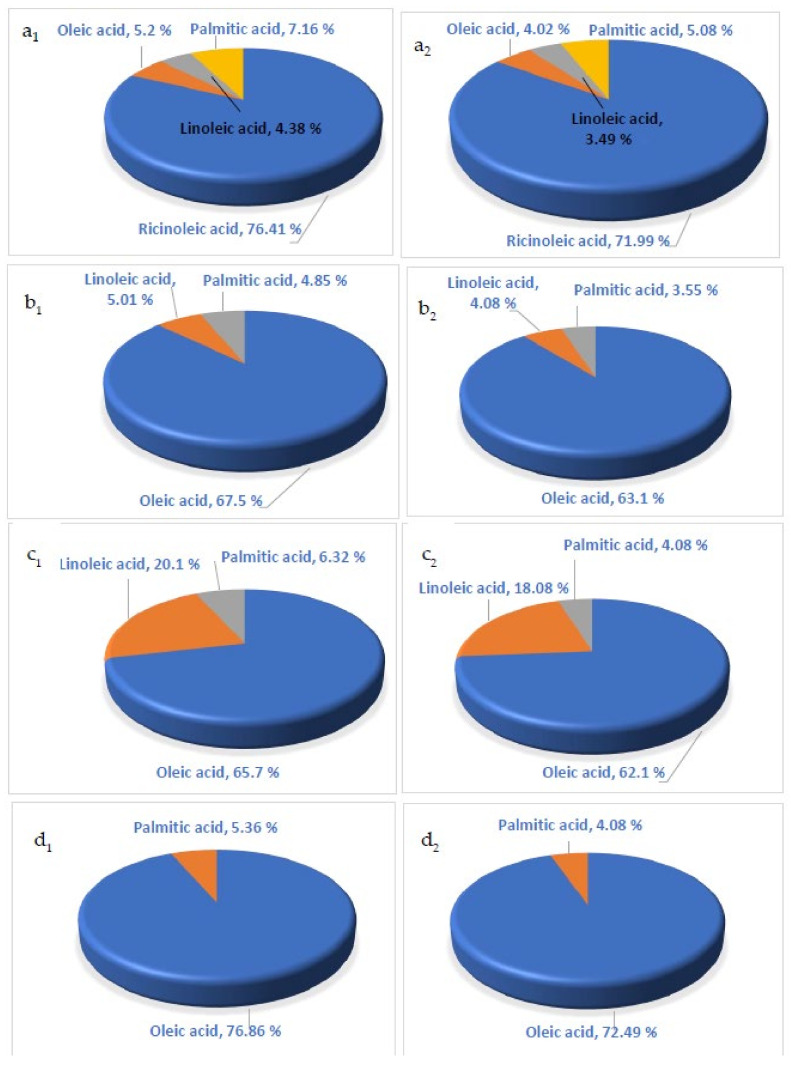
The main fatty acids and their contents of (**a_1_**,**a_2_**) Castor oil: (**a_1_**) Microwave-assisted extracted oil (MAEO), (**a_2_**) Electric hot pressed oil (EHPO), (**b_1_**,**b_2_**) Sunflower oil: (**b_1_**) For MAEO, (**b_2_**) EHPO, (**c_1_**,**c_2_**) Rapeseed oil: (**c_1_**) MAEO, (**c_2_**) EHPO, and (**d_1_**,**d_2_**) Moringa: (**d_1_**) MAEO, (**d_2_**) EHPO.

**Figure 6 polymers-15-02254-f006:**
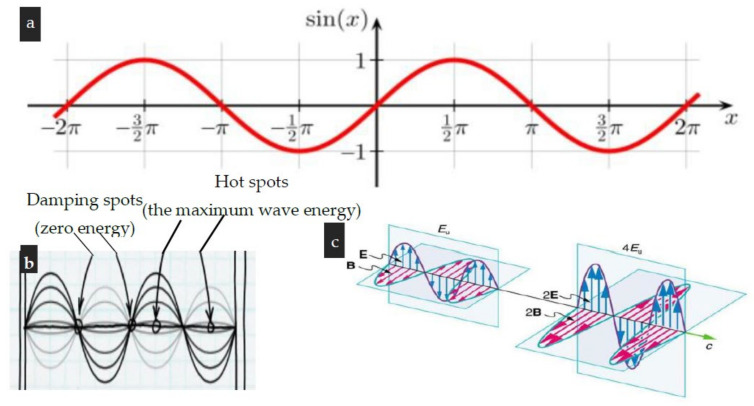
The microwave beam: (**a**) Sinusoidal wave curve (SWC), (**b**) Energy level along with the SWC, and (**c**) The proportionality between square amplitude and the energy carried by the wave.

**Figure 7 polymers-15-02254-f007:**
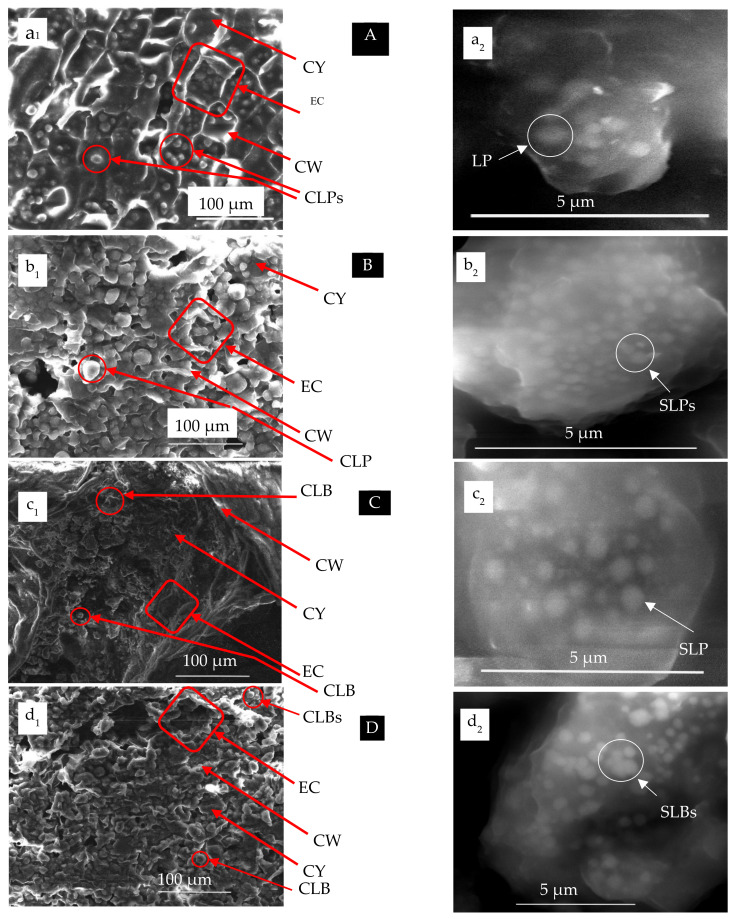
SEM micrograph of endosperm sections of the four oily seeds (**A**–**D**): (**A**) Castor bean (*Ricinus communis* L), (**B**) Common sunflower (*Helianthus annuus* L.), (**C**) Rapeseed (*Brassica napus* L.) and (**D**) Moringa (*Moringa oleifera* Lam.) showing cellular structure (**a_1_**–**d_1_**) with endosperm cells (EC), cell walls (CW), cytoplasm (CY), and compound lipid bodies (CLBs) as well as subcellular organelles (**a_2_**–**d_2_**) of singular lipid bodies (SLBs).

**Table 1 polymers-15-02254-t001:** Calculation of different chemical and physical properties of the seeds and the extracted fixed oils [7,42,48,51,52,53].

Equation	Definitions
^1^ MC, % = [(W_ads_ − W_ods_)/W_ods_] × 100 ^2^ S_cfo_, % = (W_mfo_ + W_rfo_)/W_ods_ ^3^ Y_mfo_, % = (W_mfo_/W_ods_) × 100 ^4^ Y_rfo_, % = (W_ods_ − W_rc_)/W_ods_] × 100 = (W_rfo_/W_ods_) × 100 ^5^ EL, % = [{W_ods_ − (W_rfo_ + W_rc_)}/W_ods_] × 100 ^6^ E_foe_, % = [W_rfo_/(S_cfo_ × W_ods_)] × 100	W_ads_: Weight of air-dried seeds. W_ods_: Weight of oven-dried seeds, g. W_mfo_: Weight of the main fixed oil (g). W_rfo_: Weight of recovered fixed oil (from seed’cake), g W_rc_ = Weight of residual cake after extraction. S_cfo_: Seed content of fixed oil
^7^ SG_fo_ = (W_1_ − W_2_)/(W_2_ − W_0_)	W_0_: Weight of an empty bottle. W_1_: Weight of the bottle filled with fixed oil. W_2_: Weight of the bottle filled with deionized water.
^8^ IN = 12.69 × C × (V_1_ − V_2_)/W	C, V_1_, V_2_: Parameters of sodium thiosulphate: C: Concentration. V_1_: The volume used for the blank test. V_2_: The volume used for the fixed oil. W: The fixed oil weight.
^9^ SV = 56.1N × (V_1_ − V_2_)/W	V_1_: Volume of the solution used for the blank test. V_2_: Volume of the solution used for fixed oil. N: The actual normality of the HCl used. W: The fixed oil weight.
^10^ AV = 5.61(V × N)/W	V: Volume of KOH, IN mL. N: Normality of KOH. W: Fixed oil weight.
^11^ Y_fa_ = (A_1_/A_2_) × 100	A_1_: Peak area (mm^2^) of a certain fatty acid detected at a certain retention time (min.) A_2_: The overall peak’s areas (mm^2^) of all fatty acids in the fixed oil.

^1^ Moisture content of seed, ^2^ Seed content of fixed oil, ^3^ Yield of the main fixed oil, ^4^ Yield of recovered fixed oil, ^5^ Extraction loss, ^6^ Efficiency of fixed oil extraction, ^7^ Specific gravity, ^8^ Iodine number, ^9^ Saponification value, ^10^ Acid value, ^11^ and Yield of fatty acid.

**Table 2 polymers-15-02254-t002:** Mean values * of physical and chemical properties of the four fixed oils, namely castor (C), sunflower (S), rapeseed (R), and moringa (M) extracted by microwave–hot pressing machine (MHPM) and electric hot pressing machine (EHPM).

Properties		Extraction Method ^3^								ASTM ^4^
		MHPM				EHPM				
		C	S	R	M	C	S	R	M	
MC, %	AD	±	±	±	±	±	±	±	±	
	OD	3.09 ± 0.15	±	±	±	4.47 ± 0.18	±	±	±	N.d. ^12^
OY ^5^, %		44.8 ± 1.03	42.5 ± 0.98	42.2 ± 1.02	39.6 ± 0.89	38.3 ± 1.09	36.4 ± 1.25	35.2 ± 1.33	34.6 ± 0.93	N.d. ^12^
SG ^6^		0.973 ± 0.08	0.914 ± 0.07	0.906 ± 0.1	0.946 ± 0.093	0.968 ± 0.09	0.918 ± 0.07	0.914 ± 0.09	0.957 ± 0.088	0.96–0.97
RI ^7^		1.48 ± 0.092	1.473 ± 0.08	1.465 ± 0.069	1.464 ± 0.13	1.43 ± 0.07	1.469 ± 0.086	1.467 ± 0.102	1.563 ± 0.17	1.48–1.49
IN ^8^ (g I_2_/100 g oil)		83.7 ± 1.09	126.3 ± 1.23	114.5 ± 1.48	69.7 ± 1.44	84.8 ± 1.05	127.1 ± 1.19	115.1 ± 1.29	71.4 ± 1.38	82–88
SV ^9^(mg KOH/g oil)		181.5 ± 3.79	188.4 ± 2.34	168.3 ± 1.98	180.7 ± 2.18	182.1 ± 3.07	194.6 ± 2.33	181.2 ± 2.56	190.4 ± 1.97	175–187
AV ^10^(mg KOH/g oil)		0.869 ± 0.004	0.914 ± 0.008	2.107 ± 0.08	1.433 ± 0.005	0.882 ± 0.008	1.139 ± 0.004	2.306 ± 0.006	1.344 ± 0.007	0.4–4.0
pH ^11^		6.48 ± 0.359	6.46 ± 0.45	6.43 ± 0.77	6.45 ± 0.69	6.53 ± 0.48	6.38 ± 0.39	6.49 ± 0.78	6.37 ± 0.59	N.d. ^12^

* Each value is an average of 5 samples ± standard deviation, ^3^ Yield of the main fixed oil, ^4^ Yield of recovered fixed oil, ^5^ Extraction loss, ^6^ Efficiency of fixed oil extraction, ^7^ Specific gravity, ^8^ Iodine number, ^9^ Saponification value, ^10^ Acid value, ^11^ Yield of fatty acid, and ^12^.

**Table 3 polymers-15-02254-t003:** Mean values ^1^ of lipid body (LB) diameters of the four oily seeds for each of the compound lipid body (CLP) and the single LB (SLB).

Species	Body Diameter (µm)	
	CLB	SLB
Castor bean Common sunflower Rapeseed Moringa	6.67 [0.58]–24.9 [3.25] 8.7 [0.48]–22.09 [2.74] 7.27 [0.68]–14.55 [2.06] 6.52 [0.69]–17.39 [1.83]	0.44 [0.39]–0.94 [0.03] 0.19 [0.12]–0.47 [0.05] 0.97 [0.14]–0.59 [0.08] 0.24 [0.09]–0.71 [0.09]

^1^ Each value is an average of three observations ± the standard deviations.

## Data Availability

Not applicable.

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
