# Peer review of "A Novel Microwave Hot Pressing Machine for Production of Fixed Oils from Different Biopolymeric Structured Tissues"

_polymers, 2023, doi:10.3390/polym15102254_

Round 1
Reviewer 1 Report
General comments:
In this study, a microwave hot pressing machine (MHPM) was used to heat the extruder’s colander to produce oil from each of castor, sunflower, rapeseed, and moringa seeds and compared to those obtained using ordinary electric-hot pressing machine (EHPM). The physical properties, namely oil yield (OY), specific gravity (SG), and refractive index (RI), as well as chemical properties, namely iodine number (IN), saponification value (SV), acid value (AV), pH, and chemical constituents using GC/MS of the four oils extracted by the MHPM and EHPM were determined. Chemical constituents of the resultant oil were identified using GC/MS after saponification and methylation processes. The OY and SV obtained using the MHPM were higher than those for the EHPM for all four fixed oils studied. On the other hand, each of the SG, RI, IN, AV, and pH of the fixed oils did not alter statistically due to changing the heating tool from electric band heaters into a microwave beam. The qualities of the four fixed oils extracted by the MHPM were very encouraging as a pivot of the industrial fixed oil projects compared to the EHPM. The prominent fatty acid of the castor fixed oil was found to be ricinoleic acid, making up 76.41 % and 71.99 % of the MHPM and EHPM, respectively. In addition, the oleic acid was the prominent fatty acid in each of the fixed oils of sunflower, rapeseed, and moringa species, and its yield by using the MHPM was higher than that for the EHPM. The role of microwave irradiation on facilitating fixed oil extrusion from the biopolymeric structured-organelles (lipid bodies) was protruded.
The information obtained is useful. The specific comments are as follows.
Specific comments:
1. P. 1, lines 15-16: “of to produce from” should be corrected as “to produce oil from”.
2. P. 1, lines 26-27 state “The prominent fatty acid of the castor fixed oil was found to be ricinoleic acid, making up 76.41 % and 71.99 % of the MHPM and EHPM, respectively.” It implies that 76.41 % and 71.99 % represent the content or composition. However, the caption of Figure 2 (p. 15, line 562) denotes 76.41 % and 71.99 % as the yields (Y = (W1/W2)× 100, where W1 represents the weight of the extracted fixed oil in kilograms, and W2 refers to the initial weight of the seeds (in kilograms). See p. 8, lines 305-307). Need to correct the writings of text in consistence.
3. P. 1, line 33: “thing” should be corrected as “think”.
4. P. 1, lines 31-34 state “Since it was …using microwave irradiation…more eco-friendly, more cost-effective…extraction field.”
Need to provide evidence for the claims. (Electric heating may be more
eco-friendly and cost-effective than microwave irradiation.)
5. PP. 4-6: Figures with S (for examples, line 177 Figure S1, line 195 Figure S2a-2d, line 282 Figure S6e, …) are missing.
6. P. 5, line 252: Figures 6a,d are missing.
7. P. 5, line 213: The operation conditions should be specified for MHPM and EHPM, such as power of microwave generator unit (MGU) and or temperature.
8. P. 11, Table 1: Need to specify the oil content and extraction efficiency of respective oil seeds. The oil yield is not the oil content. The oil yield divided by the oil content will give the oil extraction efficiency. The obtained oil extraction efficiency by MHPM then can be compared to those by other methods reported in literatures to show and support the excellency of MHPM claimed in this study.
9. P. 14: a2 and d2 are missing.
10. P. 15, line 562: “…their yields of…” should be corrected as “…their contents of…”.
11. P. 16, line 577, d) Palmitic acid: should indicate in which oil it was detected.
12. P. 22: Conclusion 6 is duplicated and included in Conclusion 7 and can be deleted.
13. P. 22: The statement “The primary constituent of castor oil is ricinoleic acid, making up 76.41 % and 71.99 %, of the microwave-hot presser and electric hot presser, respectively.” in Conclusion 7 is also presented in Conclusion 8 and can be omitted.
14. P. 22, Conclusion 10 seems like a suggestion rather than a conclusion.
Author Response
The Respective Reviewer 1
Notice: In this report, I mean that new numerical system belongs to my revised manuscript based on the 1st round evaluation made by the reviewers, while virgin numerical belongs to the virgin manuscript.
The 1s- modification
The nomenclature was introduced in the article just beneath the conclusion section as follow:
|
Abbreviation |
Definition |
Abbreviation |
Definition |
|
AC |
Alternate current |
MAEO |
Microwave-assisted extracted oil |
|
ACS |
The American Chemical Society |
MFT |
Maximum final temperature |
|
ADB |
Air-dried membranes |
MGU |
Microwave generator unit |
|
AFM |
Atomic force microscopy |
MHPM |
Microwave hot pressing machine |
|
ASTM |
American Society for Testing and Materials |
NDB |
Nanodehydrated-bioplastic membrane |
|
AV |
Acid value |
NIST |
Nanodehydrated-bioplastic membrane |
|
BST |
Biopolymeric Structured-Tissues |
NPS |
Nanometric particle Size |
|
C |
Castor bean (Ricinus communis L.) |
OY |
Oil yield |
|
CI |
Crystallinity index |
PD |
Pore diameter |
|
CLB |
Compound lipid bodies |
pH |
The acidity or basicity number |
|
CW |
Cell walls |
PS |
Particle size |
|
CY |
Cytoplasm |
PubChem |
An open chemistry database managed by the National Institutes of Health (NHI) |
|
DC |
Direct current |
PVA |
Polyvinyl alcohol |
|
DSC |
Differential scanning calorimetry |
R |
Rapeseed (Brassica napus L. |
|
DTA |
Differential thermal analysis |
RI |
Refractive index |
|
EC |
endosperm cells |
RT |
Residence time |
|
EHPO |
Electric hot pressed oil |
S |
Sunflower (Helianthus annuus L.) |
|
EHPM |
Electric-hot pressing machine |
SD |
Standard deviation |
|
FEG |
Field Emission Gun in the SEM |
SEM |
Scanning electron microscopy |
|
FEI |
Field Electron and Ion US-Company |
SEP |
Self-electrostatic peeling |
|
FOY |
Fixed Oil Yield |
SG |
Specific gravity |
|
FTIR |
Fourier transform infrared spectroscopy |
SLB |
Singular lipid bodies |
|
GA |
Gum Arabic |
SP |
Statistical parameters |
|
GC-MS |
Gas Chromatography-Mass spectrometer |
SR |
Surface roughness |
|
GHz |
Frequency |
SV |
Saponification value |
|
HC |
Heat change in µVs/mg |
SWC |
Sinusoidal wave curve |
|
HPM |
Hot pressing machine |
TGA |
Thermogravimetric analysis |
|
HVT |
High voltage transformer |
TR |
Temperature range |
|
IN |
Iodine number |
VFHF |
Vibrated-free horizontal flow |
|
LSD |
Least significant differencce |
VV |
Void volume |
|
M |
Moringa (Moringa oleifera Lam.) |
XRD |
X-Ray diffraction |
His general comments
1- The introduction provides sufficient background and include all relevant references.
2- All the cited references relevant to the research.
3- The methods are adequately described.
4- The conclusions supported by the results.
5- The information obtained is useful.
6- The research design can be improved: It was improved according to the reviewers 1, 2, and 3.
Moreover, the research design of the discussion section was modified that contains the following main titles:
4.1. Physical Characterization of the Fixed Oils
We deleted the following sub-titles but we retained its content as a separate paragraph:
Moisture content of seeds (MC), Fixed Oil Yield (FOY), Specific Gravity (SG), and Refractive Index (RI).
4.2. Chemical Characterization of the Fixed Oils
We deleted the following sub-titles but we retained its content as a separate paragraph:
Iodine Number (IN), Saponification Value (SV), Acid Value (AV), pH Value and GC-MS-Characterization of Fixed Oils
4.3. Effect of Microwave Irradiation on Biopolymeric Structured-Seed Tissues: was retained as it is.
4.4. Effect of Ultrastructure of seeds on Fixed Oils Extraction: was retained as it is.
7- The reviewer’ s main comment 1: The results’ presentation must be improved.
Figure 3 in page 6 (for the original numeric system) was moved to supplementary section (Figure S7). Accordingly, Figure 4 (page 17 based on the original numeric system) became Figure 3, Figure 5 (page 18 based on the original numeric system. We retained the text depending on the FigureS7 as it is without difference.
His general comments and our response are as follow:
Reviewer’ s specific comment 1:
- 1, lines 15-16 (the original numeric system): “of to produce from” should be corrected as “to produce oil from”.
Author response 1:
The statement “used to heat the colander of to produce from each of” was corrected to be “used to heat the colander to produce fixed oils from each of”.
Reviewer specific comment 2:
- 1, lines 26-27 (the original numeric system) state “The prominent fatty acid of the castor fixed oil was found to be ricinoleic acid, making up 76.41 % and 71.99 % of the MHPM and EHPM, respectively.” It implies that 76.41 % and 71.99 % represent the content or composition. However, the caption of Figure 2 (p. 15, line 562) denotes 76.41 % and 71.99 % as the yields (Y = (W1/W2)× 100, where W1 represents the weight of the extracted fixed oil in kilograms, and W2 refers to the initial weight of the seeds (in kilograms). See p. 8, lines 305-307). Need to correct the writings of text in consistence.
Author response 2:
There is a difference between: a) Oil yield [calculated by the following equation Y = (W1/W2) × 100, where W1 represents the weight of the extracted fixed oil in kilograms, and W2 refers to the initial weight of the seeds (in kilograms)] and fatty acid yield, and its values are presented in Table 1.
- b) Fatty acid yield (estimated from the peak area of the GC/MS diffractogram), and its values are presented in Figure 2.
The values referred by the respective reviewer are correct, but I agree with him in well-defining the difference between the oil yield (above-mentioned equation) and the fatty acid yield (estimated from the peak area of a certain fatty acid through the GC/MS chromatogram).
To differentiate the difference between oil yield (OY) and the fatty acid, I have added the phrase “content of oil” to the fatty acid to become “fatty acid yield” and, moreover, I abbreviated it as “FAY”. In addition, I protruded this concept in the materials and methods section.
Accordingly, to prevent this confusion of concepts, I have corrected this in the text as the respective reviewer referred and commented.
-- In page 1, line 27 (in the abstract section based on the original numeric system), I have written the following statement: “, making up 76.41 % and 71.99 % contents of the oils extracted by the MHPM and EHPM, respectively” instead of “making up 76.41 % and 71.99 % of the MHPM and EHPM, respectively”.
-- In the Materials and Methods section: we modified and added statements and a formula to treat this valuable remark in Page 10 in a new lines (411-417) as follow:
A peak area normalization method was also used to determine the fatty acid yield (FAY) of the chemical constituents of the oil [54] according to the following formula: Y = (A1/A2) ×100, where W1 represents the peak area (in mm2) of a certain fatty acid detected at a certain retention time (in minutes), and W2 refers to the overall areas in mm2 representing chemical constituents of the fixed oil that arisen at different retention times (in minutes) at the chromatogram.
In addition, the following references was added and the numbers’ of the references were readjusted in a correct sequence: “61-Gaul, J.A. Quantitative Calculation of Gas Chromatographic Peaks in Pesticide Residue Analyses. J. Assoc. Off. Anal. Chem. 1966, 49, 389-399”.
Reviewer specific comment 3:
- 1, line 33: “thing” based on the original numeric system should be corrected as “think”.
Author response 3:
“thing” was corrected to be “think”.
Reviewer specific comment 4:
- 1, lines 31-34 based on the original numeric system state “Since it was …using microwave irradiation…more eco-friendly, more cost-effective…extraction field.”
Need to provide evidence for the claims. (Electric heating may be more eco-friendly and cost-effective than microwave irradiation.)
Author response 4:
To show our defense on the concluded claims, the following paragraphs confirm the truth of these claims: against the following comment of the respective reviewer found in the abstract section “Since it was confirmed by the present study that using microwave irradiation is simple, facile, more eco-friendly, cost-effective, retains parent quality of oils, and allows to worm bigger machines and spaces, we think it will make an industrial revolution in oil extraction field”
- The following paragraph (Pages 3, 4, lines 150-153) “Contrarily, microwave offers a thermally-homogeneous spaces in any desired size. In addition, the ordinary available electric hot presser machines depend for their heat transferring, mainly on the conduction phenomenon that increases the oil extraction durations and reduces the extraction efficiency itself [….]”.
- In page 16, lines 538-541, “Microwave beams did not alter the parent oil quality comparing to other ordinary methods [43]. As a result of the present investigation, the major advantage of microwave treatment is the reduced time of extraction and energy consumption costs comparing to ordinary methods. Analysis of oil quality indices that oils from both processes (conventional and microwave assisted-extractions) have comparatively similar properties. Accordingly, it was justification of microwaves use as a rapid tool for oil extraction process [43,44]”.
- In the results section (page 16, lines 538-541): “At the same level of oily species, the MHPM produces nearly identical values for those obtained by the EHPM (Table 2). This indicates that microwave irradiation did not alter the SGfo of the resultant fixed oils protruding the economical importance of this novel investigation”.
- In the results section (page 17, lines 551-553): “This indicates that microwave beams did not cause any difference for the RI of the resultant fixed oils. This results confirms the suitability of the present study for the present industrial application”.
- In page 23, lines 711-714: “Since the MHPM produces nearly identical values for those obtained by the EHPM at the same level of oily species (Table 1), it can be concluded that microwave beams did not cause any difference for the RI of the resultant fixed oils. This results confirms the suitability of the present study for the present industrial application”.
- In page 25, lines 765-774: “In microwave heating, microwaves penetrate to the interior of biopolymeric structured-tissues including but not limited to seeds as well as foods affecting their dipolar molecules of water, fats, or sugars. Accordingly, heat is generated because of dipolar molecules absorb electromagnetic radiation and, leading to their intensive vibration producing friction that leads to rapid temperature rising and consequently efficient water evaporation and/or melting fats. This results in a greatly increased vapor pressure differential between the center and surface of the biopolymeric tissue, allowing fast transfer of moisture and or essential oil out of the tissue. Hence, microwave drying as well as microwave hot pressing technique is rapid, more uniform and energy efficient compared to ordinary ones [48,84,85]”.
- In page 25, lines 799-801: Accordingly, using the microwave hot pressing machine exhibits an ideal tool for heat transfer and subsequently heat the oily seed tissues and increases the fixed oil yield [84-86].
- In page 26, lines 840-848: Comparing with conventional methods, microwave treatment for oil extraction has many advantages e.g. improvement of extracted oil yield and quality, direct extraction capability, lower energy consumption, faster processing time and reduced solvent contents [7,80]. These results may be attributed to that microwave irradiation provides a potential alternative to induce stress reactions in ultrastructured-tissues within the oil seeds. By using microwave radiation in oil seeds, a higher extraction yield and an increase in mass transfer coefficients can be obtained because the cell membrane is more severely ruptured. Apart from this, permanent pores were generated as accordingly, this enables the oil to move through permeable cell walls [42,44].
Reviewer specific comment 5:
- PP. 4-6: Figures based on the original numeric system with S (for examples, line 177 Figure S1, line 195 Figure S2a-2d, line 282 Figure S6e
Author response 5:
Reviewer specific comment 6:
- 5, line 252 based on the original numeric system: Figures 6a,d are missing.
Author response 6:
There is a mistake was corrected. “Figure 6a,d” was corrected to be “Figure S6a,d” in page 6, line 252 (the new sequence).
Reviewer specific comment 7:
- 5, line 213 based on the original numeric system: The operation conditions should be specified for MHPM and EHPM, such as power of microwave generator unit (MGU) and or temperature.
Author response 7:
The operation conditions including temperature were already have written in the Material and Methods section as follow:
In page 5, lines 217-218 (based on the new arisen numeric system): The main components of the HPM are the Taby motor (380 Volt 3 phase 10 amp 50/60 Hz, Sweden) ……
In page 5, lines 239-255 based on the new arisen numeric system: In the MGU, the magnetron is the primary and only source of microwave energy (Figure S6a,d). Originally, it was taken from a domestic microwave oven. In terms of technical specifications, the magnetron has a power of 900W, an anode voltage of 4.20 kVp, and a frequency of 2.46 GHz. The magnetron, 2M214 39F (06B), code of 2B71732E for LG microwave ovens (900 W, 4.20 kVp anode voltage, 2460 MHz frequency) was used in this study as a self-excited microwave oscillator, cavity magnetrons convert high-voltage electric energy into microwave beams (Figure S6a,d).
It is worth for mentioning that magnetrons produce high-power outputs by crossing electron and magnetic fields. In this study, a magnetron emits a constant frequency of 2.46 GHz, since magnetrons are usually designed on a constructively fixed frequency as illustrated by Hindi et al. [46-48].
As shown in Figure 5b, microwave energy is radiated into the air through a stain-less steel pipe with a diameter of 1 inch, then is conducted into a metal cavity. In order to melt the fixed oil and forced it to excrete from seed tissues, temperature must be adjusted to be approximately 75-80°C.
Reviewer specific comment 8:
- 11, Table 1: Need to specify the oil content and extraction efficiency of respective oil seeds. The oil yield is not the oil content. The oil yield divided by the oil content will give the oil extraction efficiency. The obtained oil extraction efficiency by MHPM then can be compared to those by other methods reported in literatures to show and support the excellency of MHPM claimed in this study.
Author response 8:
This comment of the reviewer is very logic and scientifically appreciated by us.
We added the titles of Oil Extraction Efficiency (OEE) in Material and Methods bearing no. 2.3.1.4, Results section bearing no. 3.1.4. as well as in discussion section as a untitled paragraph.
In conclusion: The tenth paragraph in conclusion was modified to be ” The oil extraction efficiency showed that using microwave irradiation enhanced the oil yield with retaining the parent quality of the fixed oils that is very encouraging for candidating such an invention as a pivot of the industrial fixed oil projects ”.
Reviewer specific comment 9:
- 14: a2 and d2 are missing.
Author response 9:
In Figure 2: a2 was rewrote and d2 was adjusted
Reviewer specific comment 10:
- 15, line 562 based on the original numeric system: “…their yields of…” should be corrected as “…their contents of…”.
Author response 10:
In page 16, line 577 (new numeric system) in the title of Figure 2 we corrected as follow: The main fatty acids and their contents
Reviewer specific comment 11:
- P. 16, line 577 based on the original numeric system, d) Palmitic acid: should indicate in which oil it was detected.
Author response 11:
The comment was considered by adding: Palmitic acid detected at the four fixed oils in the supplementary material page 8.
Reviewer specific comment 12:
- 22: Conclusion 6 is duplicated and included in Conclusion 7 and can be deleted.
Author response 12:
In conclusion, the point no. 6 was deleted, and the followed points were re-numbered in a correct sequence.
Reviewer specific comment 13:
- 22 based on the original numeric system: The statement “The primary constituent of castor oil is ricinoleic acid, making up 76.41 % and 71.99 %, of the microwave-hot presser and electric hot presser, respectively.” in Conclusion 7 is also presented in Conclusion 8 and can be omitted.
Author response 13:
The paragraph no.7 was deleted from the conclusion section.
Reviewer specific comment 14:
- 22 based on the original numeric system, Conclusion 10 seems like a suggestion rather than a conclusion.
Author response 14:
The tenth paragraph in conclusion was modified to be ” The oil extraction efficiency showed that using microwave irradiation enhanced the oil yield with retaining the parent quality of the fixed oils that is very encouraging for candidating such an invention as a pivot of the industrial fixed oil projects ”.
Moreover, the statement in conclusion, page 22, lines 788-789 (original numerical system) “....tools were determined, compared together, and evaluated based on those in literature for the four fixed oils” was modified to be “tools were evaluated based on those in literature for the four fixed oils”.
Four paragraphs (no’s 5, 6, 7, 10) were deleted, and, subsequently.
Further Modifications:
4.2.5. GC-MS-Characterization of Fixed Oils
These fatty acids are represented as methyl or methyl esters due to pretreatments of the fixed oils using the saponification/ methylation route. It has been demonstrated that methylation of oil causes the oil to become more volatile, which subsequently adapts the Programme Temperature Volume of the injector of the GC-MS device [65,66]. There is no difference in fragmentation patterns between methyl esters derivatives of free fatty acids and their parent free fatty acids, except that methyl esters have a higher molecular weight than their parent free fatty acids by 14 grams/moles [66].
Examining Figure 2 concerning to the four fixed oils resulted by the MHPM and EHPM revealed that four fatty acids, namely ricinoleic, oleic, linoleic and palmitic acids were detected, but in different concentrations (Figure 2, S7a-d).
Comparisons of the squeezed fixed oils were performed within and between species to ensure that novel procedures used did not distort their parent properties examined.
Comparisons between species revealed that castor bean had the most abundant ricinoleic acid among the four species studied, while moringa had the highest content of oleic acid. For linoleic acid, it was detected only in castor beans, sunflower and rapeseed, while moringa was found to be free of it. Palmetic acid was detected at all the four species in which castor bean and rapeseed had higher contents of it compared than those for sunflower and moringa. Moreover, comparisons within species indicated that the fixed oils obtained by MHPM gave higher contents of the fatty acids than those produced by EHPM.
Since castor oil is a unique source of ricinoleic acid, a hydroxylated fatty acid [9,47,69], high content of ricinoliec acid can cause the oil to be converted from a non-drying oil to a drying oil by removing the hydroxyl groups in its chain. This is useful in the manufacture of alkyd resin coating [66] and adds to its industrial advantages. The presence of palmitic acid in castor oil may has an economic value to its sodium salt, which is used in the soap and cosmetic industries [52,67].
Effect of microwave on oil extraction
The major advantage of microwave is the reduced time of extraction and energy consumption costs comparing to ordinary techniqyes.
Since the novel MHPM procedure used did not distort parent oil properties examined giving comparatively similar properties, thus it can be justified that using microwaves irradiation is considered as a rapid tool for oil extraction [43].
Enhancing yield and quality of fixed oils by employing microwave irradiation as a heating technique will help industry to more growing and offers mass production due to possibility of worming bigger machines and spaces and fasten production rate [43, 46-48].
The newly introduced references
Seventeen references were added to fill any gap in the manuscript according to the respective reviewers’ comments:
- Farag, R. S.; Hewedi, F.M.; Abu-Raiia, S.H.; Elbaroty, G.S. Comparative study on the deterioration of oils by microwave and conventional heating. Food Prot. 1992, 55: 722-727.
- Takagi, S.; Ienaga, H.; Tsuchiya, C.; Yoshida, H. Microwave roasting effects on the composition of tocopherols and acyl lipids within each structural part and section of a soya bean. Sci. Food Agric. 1999a, 79: 1155-1162.
- Takagi, S.; Yoshida, H.. Microwave heating influences on fatty acid distribution of triacylglycerols and phospholipids in hypocotyls of soybeans (glycine max). Food Chem. 1999b, 66, 345–351.
- Ramanadhan, B. Microwave extraction of essential oils (from black pepper and coriander) at 2.46 Ghz, Canada: University of Saskatchewan, MSc thesis. 2005.
- Uquiche, E.; Jerez, M.; Ortiz, J. Effect of pretreatment with microwaves on mechanical extraction yield and quality of vegetable oil from Chilean hazelnuts (Gevuina avellana Mol). Innov. Food Sci. Emerg. Technol. 2008, 9, 495–500.
- Yoshida, H.; Hirakawa, Y.; Tomiyama, Y.; Mizushina, Y. Effects of microwave treatment on the oxidative stability of peanut (Arachis hypogaea) oils and the molecular species of their triacylglycerols. Eur. Lipid Sci. Technol, 2003, 105, 351-358.
- G.W.; Robertson, J.A. Moisture content/relative humidity equilibrium of high-oil and confectionery type sunflower seed. J. Stored Prod. Res. 1987, 23, 115-118.
- Doan, L.G. Ricin: mechanism of toxicity, clinical manifestations, and vaccine development, A Review.J. Toxicol. 2004, 42, 201– 208.
- Akaranta, O.; Anusiem, A.C.I. A boiresource solvent for extraction of castor oil. Ind Crops Prod. 1996, 5, 273-277.
- Gaul, J.A. Quantitative Calculation of Gas Chromatographic Peaks in Pesticide Residue Analyses. Assoc. Off. Anal. Chem. 1966, 49, 389-399
- Doan, L.G. Ricin: mechanism of toxicity, clinical manifestations, and vaccine development, A Review.J. Toxicol. 2004, 42, 201– 208.
- Akaranta, O.; Anusiem, A.C.I. A boiresource solvent for extraction of castor oil. Ind Crops Prod. 1996, 5, 273-277.
- Hansel, F.A; Bull, I.D.; Evershed, R.P. Gas chromatographic mass spectrometric detection of dihydroxy fatty acids preserved in the ‘bound’ phase of organic residues of archaeological pottery vessels. Rapid Commun. Mass Spectrom. 2011, 25, 1893–1898. Hansel
- Matthaus, B.; Özcan, M.; Al Juhaimi, F. Some rape/canola seed oils: fatty acid composition and tocopherols. Zeitschrift für Naturforschung C. 2016, 71, 73-77. Matthaus et al, 2016
- Chew, S.C. Cold pressed rapeseed (Brassica napus) oil. In Green Technology, Bioactive Compounds, Functionality, and Applications. M. F. Eds., Academic Press. 2020, 65-80. Chew, 2020
- Doan, L.G. Ricin: mechanism of toxicity, clinical manifestations, and vaccine development, A Review.J. Toxicol. 2004, 42, 201– 208. Doan 2004
- Akaranta, O.; Anusiem, A.C.I. A boiresource solvent for extraction of castor oil. Ind Crops Prod. 1996, 5, 273-277. Akaranta, O.; Anusiem 1996.
- Microwave Assisted Drying. 1993. http://ecoursesonline.iasri.res.in/mod/page/view.php?id=882 (accessed on 4 April 2023).
- Rakesh, V.; Seo, Y.; Datta, A. K.; McCarthy, K. L.; McCarthy, M. J. Heat transfer during microwave combination heating: Computational modeling and MRI experiments. Food Nat. Prod. 2010, 56, 2468-2478.
- Watanabe, S.; Karakawa, M.; Hashimoto, O. Temperature of a heated material in a microwave oven considering change of complex relative permittivity. European Microwave Conference (EuMC), Rome, Italy. 2009, 798-801.
The design of materials and methods, results, and discussion sections were enhanced by dividing into the following titles and subtitles:
For the Materials and Methods: The sequence of the main and subtitles are as follow:
- Materials and Methods
2.1. Management plan
2.2. Raw Materials
2.2.1. Preparation of the seeds
2.2.2. Extraction of the Four Fixed Oils
2.2.3. The Hot Pressing Machine (HPM)
2.3. Characterization of the Four Species
2.3.1. Physical Characterization of the Seeds and Oils
2.3.2. Chemical Characterization of the Fixed oils
2.3.2.1. Determination of Iodine Number (IN)
2.3.2.2. Determination of Saponification Value (SV)
2.3.2.3. Determination of Acid Value (AV)
2.3.2.4. Determination of pH Value
2.3.2.5. Determination of the Fatty Acids of the Fixed Oils
- Preparation of the Methyl Esters
- Gas Chromatography-Mass spectrometer (GC–MS)
2.4. Scanning Electron Microscopy (SEM)
2.5. Statistical Design and Analysis
Furthermore, all the equations used to calculate the mean values of the properties of the seeds as well as the fixed oils were collected and presented in a new Table numbered as Table 1 as follow:
Table 1. Calculation of different chemical and physical properties of the seeds and the extracted fixed oils
[7,48,51-53,85].
|
Equation |
Definitions |
|
|
1 MC, % = [(Wads-Wods)/Wods] × 100 2 Scfo, % = (Wmfo+Wrfo)/Wods 3 Ymfo, %= (Wmfo/Wods) ×100 4 Yrfo, % = (Wods-Wrc)/Wods] ×100= (Wrfo / Wods) ×100 5 EL, % = [{Wods-(Wrfo+Wrc)}/Wods]×100 6 Efoe, % = [Wrfo/(Scfo ×Wods)]×100 |
Wads: Weight of air- dried seeds. Wods: Weight of oven-dried seeds, g. Wmfo: Weight of the main fixed oil (g). Wrfo: Weight of recovered fixed oil (from seed’cake), g Wrc = Weight of residual cake after extraction. Scfo: Seed content of fixed oil |
|
|
7 SGfo = (W1-W2)/( W2-W0) |
W0: Weight of an empty bottle. W1: Weight of the bottle filled with fixed oil. W2: Weight of the bottle filled with deionized water. |
|
|
8 IN = 12.69×C × (V1-V2)/W |
C, V1, V2: Parameters of sodium thiosulphate: C: Concentration. V1: The volume used for the blank test. V2: The volume used for the fixed oil. W: The fixed oil weight. |
|
|
9 SV = 56.1N × (V1-V2)/W |
V1: Volume of the solution used for the blank test. V2: Volume of the solution used for fixed oil. N: The actual normality of the HCl used. W: The fixed oil weight. |
|
|
10 AV=5.61(V×N)/W |
V: Volume of KOH, IN mL. N: Normality of KOH. W: Fixed oil weight. |
|
|
11 Yfa = (A1/A2)×100 |
A1: Peak area (mm2) of a certain fatty acid detected at a certain retention time (min.) A2: The overall peak’s areas (mm2) of all fatty acids in the fixed oil. |
|
1 Moisture content of seed, 2 Seed content of fixed oil, 3 Yield of the main fixed oil , 4 Yield of recovered fixed oil, 5 xtraction loss, 6 Efficiency of fixed oil extraction, 7 Specific gravity, 8 Iodine number, 9 Saponification value, 10 Acid value, 11 Yield of fatty acid.
For the results section: The sequence of the main and subtitles are as follow:
- Results
3.1. Physical Properties of seeds
Two paragraphs were included under this title belonging to:
- Moisture content of seeds (MCs), and
- The seed content of fixed oil (Scfo).
3.2. Physical Properties of the Fixed Oils
Without subtitles, six paragraphs were covered this title illustrating each of the following fixed oil properties:
- Yield of the main fixed oil (Ymfo)
- Yield of recovered fixed oil (Yrfo)
- Extraction loss (EL)
- Efficiency of the fixed oil extraction (Efoe)
- Specific gravity of fixed oil (SGfo)
- Refractive Index (RI)
3.3. Chemical Properties of the Fixed Oils
Also, without subtitles, five paragraphs were covered this title illustrating each of the following fixed oil properties:
- Iodine Number (IN)
- Saponification Value (SV)
- Acid Value (AV)
- pH Value
- Yield of fatty acid (Yfa).
3.4. Effect of Microwave Irradiation on Biopolymeric Structured-Tissues (BST): The title was retained without changing.
For the discussion section: The sequence of the main and subtitles are as follow:
- Discussion
4.1. Physical Properties of seeds
Two paragraphs were included under this title belonging to:
- Moisture content of seeds (MCs), and
- The seed content of fixed oil (Scfo).
4.2. Physical Properties of the Fixed Oils
Without subtitles, six paragraphs were covered this title illustrating each of the following fixed oil properties:
- Yield of the main fixed oil (Ymfo)
- Yield of recovered fixed oil (Yrfo)
- Extraction loss (EL)
- Efficiency of the fixed oil extraction (Efoe)
- Specific gravity of fixed oil (SGfo)
- Refractive Index (RI)
4.3. Chemical Properties of the Fixed Oils
Also, without subtitles, five paragraphs were covered this title illustrating each of the following fixed oil properties:
- Iodine Number (IN)
- Saponification Value (SV)
- Acid Value (AV)
- pH Value
- Yield of fatty acid (Yfa).
These old subtitles with their numbers was removed, while their text remained without changing.
4.4. Effect of Microwave Irradiation on Biopolymeric Structured-Tissues (BST): The no. of the title was changed, while its text remained without changing.
4.5. Effect of Ultrastructure of seeds on Fixed Oils Extraction
Also, the no. of the title was changed, while its text remained without changing.

Reviewer 2 Report
This article compares the oil extracted from four oilseeds by two methods based on hot extrusion. The fundamental difference between the two methods is that the heating is carried out in one of them by means of a conventional electrical resistance and the other by microwaves. In general, the characterization made of the oils obtained is limited, only some physical indices and their composition in fatty acids have been determined.
In the results section, it seems unnecessary to include the mass spectra of the different fatty acids, since these are known and should not present identification problems. On the other hand, although the study carried out by SEM is interesting, it is focused only on the original seeds and the relationship it has with the extraction methods is not very clear.
The English Language needs a review.
Line 53: Probably you meant “tissues” instead of “tisuues”.
Line 69: Please change “moriga” by “moringa”.
Line 72-78: This paragraph seems out of place.
Line 76: Please delete one of the parenthesis.
Line 76: Add a “.” Before Accordingly.
Line 164-165: Please revise this sentence “In addition, temperature distribution along with the oil-extraction machine is heterogeneous that related to the heated zones by the electric tools.”
Line 173-174: Please revise this sentence “If these results will be realized, using microwave irradiation as a heating tool through oil extraction will be industrially-recognized”
Line 214: Please revise the sentence “Microwave radiation is generated using the MGU by converting alternate electric current (AC) into microwave beam that responsible on worming the HPM.”
Line 220: Probably you meant “necessary to warm” instead of “nessecary to worm”
Line 269: Is there anything missing after “In addition,”?
Line 343: Please specify the volume of ethanol.
Line 370: the heading of the section “Determination of the Chemical Constituents of the Fixed Oils” should be “Fatty acids determination” or something similar since no other components are analysed.
Line 378: The chromatography in silica gel column should be describe.
Line 395: Which other components? Apparently you have analyse only fatty acids.
Line 464: “For the SG property, there are no statistica differences between the SG values among the four species examined, else the comparisons made within or between species.” I´m sorry but don´t understand this sentence.
Line 464-472: In the paragraph, If you say at the beginning that there are no statistical differences, then how can you say that some are greater or smaller than others?
Figure 2: a2 is missing and d2 is only “d”
Author Response
The Respective Reviewer 2
Notice: In this report, I mean that new numerical system belongs to my revised manuscript based on the 1st round evaluation made by the reviewers, while virgin numerical belongs to the virgin manuscript.
The 1s- modification
The nomenclature was introduced in the article just beneath the conclusion section as follow:
|
Abbreviation |
Definition |
Abbreviation |
Definition |
|
AC |
Alternate current |
MAEO |
Microwave-assisted extracted oil |
|
ACS |
The American Chemical Society |
MFT |
Maximum final temperature |
|
ADB |
Air-dried membranes |
MGU |
Microwave generator unit |
|
AFM |
Atomic force microscopy |
MHPM |
Microwave hot pressing machine |
|
ASTM |
American Society for Testing and Materials |
NDB |
Nanodehydrated-bioplastic membrane |
|
AV |
Acid value |
NIST |
Nanodehydrated-bioplastic membrane |
|
BST |
Biopolymeric Structured-Tissues |
NPS |
Nanometric particle Size |
|
C |
Castor bean (Ricinus communis L.) |
OY |
Oil yield |
|
CI |
Crystallinity index |
PD |
Pore diameter |
|
CLB |
Compound lipid bodies |
pH |
The acidity or basicity number |
|
CW |
Cell walls |
PS |
Particle size |
|
CY |
Cytoplasm |
PubChem |
An open chemistry database managed by the National Institutes of Health (NHI) |
|
DC |
Direct current |
PVA |
Polyvinyl alcohol |
|
DSC |
Differential scanning calorimetry |
R |
Rapeseed (Brassica napus L. |
|
DTA |
Differential thermal analysis |
RI |
Refractive index |
|
EC |
endosperm cells |
RT |
Residence time |
|
EHPO |
Electric hot pressed oil |
S |
Sunflower (Helianthus annuus L.) |
|
EHPM |
Electric-hot pressing machine |
SD |
Standard deviation |
|
FEG |
Field Emission Gun in the SEM |
SEM |
Scanning electron microscopy |
|
FEI |
Field Electron and Ion US-Company |
SEP |
Self-electrostatic peeling |
|
FOY |
Fixed Oil Yield |
SG |
Specific gravity |
|
FTIR |
Fourier transform infrared spectroscopy |
SLB |
Singular lipid bodies |
|
GA |
Gum Arabic |
SP |
Statistical parameters |
|
GC-MS |
Gas Chromatography-Mass spectrometer |
SR |
Surface roughness |
|
GHz |
Frequency |
SV |
Saponification value |
|
HC |
Heat change in µVs/mg |
SWC |
Sinusoidal wave curve |
|
HPM |
Hot pressing machine |
TGA |
Thermogravimetric analysis |
|
HVT |
High voltage transformer |
TR |
Temperature range |
|
IN |
Iodine number |
VFHF |
Vibrated-free horizontal flow |
|
LSD |
Least significant differencce |
VV |
Void volume |
|
M |
Moringa (Moringa oleifera Lam.) |
XRD |
X-Ray diffraction |
His general comments
1- The introduction provides sufficient background and include all relevant references.
2- All the cited references relevant to the research.
3- The methods can be improved: It was improved according to the reviewers 1, 2 , and 3.
4- The conclusions must be improved: It was improved.
5- The research design must be improved: It was improved according to the reviewers 1, 2, and 3.
6- The results’ presentation must be improved: It was improved.
His comments and our response are as follow:
Reviewer comment 1:
This article compares the oil extracted from four oilseeds by two methods based on hot extrusion. The fundamental difference between the two methods is that the heating is carried out in one of them by means of a conventional electrical resistance and the other by microwaves. In general, the characterization made of the oils obtained is limited, only some physical indices and their composition in fatty acids have been determined.
Author response 1:
We added additional characterizations based on the comments of the reviewers to enhance the arisen impressions concerning to the net efficiency of the extraction techniques as related by moisture content as well as the other performed characterizations of the seeds in order to confirm that the microwave enhanced the oil quality as well as keeping parent quality of the oil:
- Physical properties: Moisture content (MC), oil content (OC), oil yield (OY), oil extraction efficiency (OEE), specific gravity (SG), and refractive index (RI).
- Chemical properties: iodine number (IN), saponification value (SV), acid value (AV), pH, fatty acids.
Reviewer comment 2:
In the results section, it seems unnecessary to include the mass spectra of the different fatty acids, since these are known and should not present identification problems. On the other hand, although the study carried out by SEM is interesting, it is focused only on the original seeds and the relationship it has with the extraction methods is not very clear.
Author response 2:
Regarding to the respective reviewer 2 commented about unnecessary presence of mass spectra of the different fatty acids, we have moved Figure 3 from the main article into supplementary (Figure S7). The presence of this Figure in the supplementary can help readers to study its content without searching efforts about, else chemical constituent or mass spectra of the analyzed fatty acids.
Reviewer comment 3:
The English Language needs a review.
Author response 3: English language of the manuscript will be revised by the famous MDPI-Editing Office automatically after acceptance.
Reviewer comment 4:
Line 53 based on the original numeric system: Probably you meant “tissues” instead of “tisuues”.
Author response 4:
Yes, I corrected tisuues into tissues at page 2, line 53.
Reviewer comment 5:
Line 69 based on the original numeric system: Please change “moriga” by “moringa”.
Author response 5:
The word moriga was corrected into moringa
Reviewer comment 6:
Line 72-78 based on the original numeric system: This paragraph seems out of place.
Author response 6:
The referred paragraph was deleted, while some of its statements were moved and distributed at other sections such as discussion.
Reviewer comment 7:
Line 76 based on the original numeric system: Please delete one of the parenthesis.
Author response 7:
One of the duplicated brackets was deleted.
Reviewer comment 8:
Line 76 based on the original numeric system: Add a “.” Before Accordingly.
Author response 8:
“.” before Accordingly was added.
Reviewer comment 9:
Line 164-165 based on the original numeric system: Please revise this sentence “In addition, temperature distribution along with the oil-extraction machine is heterogeneous that related to the heated zones by the electric tools.”
Author response 9:
To make the sentence “In addition, temperature distribution along with the oil-extraction machine is heterogeneous that related to the heated zones by the electric tools” more clearance, it was modified to be “Contrarily, microwave offers a thermally-homogeneous spaces in any desired size” (page, lines:)
Reviewer comment 10:
Page 4/Line 173-174 based on the original numeric system: Please revise this sentence “If these results will be realized, using microwave irradiation as a heating tool through oil extraction will be industrially-recognized”
Author response 10:
We changed the statement “It was expected that yield and quality of all the four fixed oils will be either constant or enhanced due to using microwave. If these results will be realized, using microwave irradiation as a heating tool through oil extraction will be industrially-recognized into the following statement: “Enhancing yield and quality of fixed oils by employing microwave irradiation as a heating technique will help industry to more growing due to possibility of worming bigger machines and spaces”.
Reviewer comment 11:
Page 5 Line 214 based on the original numeric system: Please revise the sentence “Microwave radiation is generated using the MGU by converting alternate electric current (AC) into microwave beam that responsible on worming the HPM”.
Author response 11:
We changed the sentence of “Microwave radiation is generated using the MGU by converting alternate electric current (AC) into microwave beam that responsible on worming the HPM” into the next one to be more clear: “Microwave radiation is generated using the magnetron device, the essential component of the MGU, by converting the input alternate electric current (AC) into microwave beam that responsible on worming the HPM” (Page 5, Lines: 214-216).
Reviewer comment 12:
Line 220 based on the original numeric system: Probably you meant “necessary to warm” instead of “necessary to worm”
Author response 12:
Page 5 - Line 220 based on the original numeric system: We changed “necessary to warm” into “necessary to worm”
Reviewer comment 13:
Line 269: Is there anything missing after “In addition,”?
“In addition,…..The high voltage” into “In addition, the high voltage”
Author response 13:
The above-mentioned comment was considered and corrected as shown in page 5 and line 255 (new numeric system).
Reviewer comment 14:
Line 343 based on the original numeric system: Please specify the volume of ethanol.
Author response 14:
We changed the following paragraph:
“About 0.5 g of dissolved in ethanol was mixed with 10 ml of potassium hydroxide (2.5 N) over a period of two hours, then cooled. An oil-free KOH solution is used as a blank solution. The unreacted portion of the KOH in either the oil sample or blank solution was titrated with oxalic acid (2.5 N) using phenolphthalein as an indicator. Based on the following equation, we were able to determine the saponification SV value.
SV=[56×(V1-V2)×1000]/[2×1000×W], where W refers to the weight of the castor oil (g), V1 refers to the volume of 0.5N oxalic acid for the blank solution (mL), and V2 refers to the volume of 0.5 N oxalic acid for the fixed oil (mL)” into the next paragraph (Page 9, lines 343-349, yellow-highlighted):
About 2 g of fixed oil was added to 25 ml of 0.1 N ethanolic potassium hydroxide with constant stirring was allowed to be boiled gently for 60 min under refluxing. Few drops of phenolphthalein indicator was added to the warm mixture and then titrated with HCl (0.5 M) up to the end point in which the pink color of the indicator just disappeared. Based on the following equation, we were able to determine the saponification SV value.
S.V = 56.1N × (V1-V2)/M, where V1 = the volume of the solution used for blank test (mL), V2 = the volume of the solution used for determination (mL), N = Actual normality of the HCl used, and M = Mass of the fixed oil sample.
Reviewer comment 15:
Line 370: the heading of the section “Determination of the Chemical Constituents of the Fixed Oils” should be “Fatty acids determination” or something similar since no other components are analysed.
Author response 15:
The title “2.3.2.5. Determination of the Chemical Constituents of the Fixed Oils” was changed to “Fatty acids determination”.
Reviewer comment 16:
Line 378 based on the original numeric system: The chromatography in silica gel column should be describe.
Author response 16:
Some essential information were added as follow in page, lines 379-395 based on the original numeric system:
“The procedure includes the following steps: saponification, esterification, fractionation on a column chromatography filled with silica gel, and analysis by GC/MS. To remove the free lipids, about 2 g of fixed oil was extracted with 10 mL mixture of CHCl3/MeOH solution (2:1 v/v) and sonicated for about 15 min, each. To release "bound" components, the free lipids-fixed oil sample was saponified with 0.5 M NaOH (MeOH/H2O solution, 9:1 v/v, 5 mL, 70 °C, 1 h).
After centrifugation of the cold mixture at 2500 rpm for 15 minutes, the supernatant was discarded, the precipitate was collected and its pH was readjusted to be 3 by adding 3 M HCl. About 1 mL of solvent-extracted water, and 3 mL of CHCl3 were used to extract the hydrolyzed lipids. After complete elimination of the solvent by evaporation under a steady flowing of nitrogen atmosphere, the hydrolyzed acid components were methylated using 100 L of BF3/MeOH (14% w/v) at 70 °C for 1 h). Then, the methyl esters were extracted with about 2 mL of CHCl3 (3 times) from the solvent-extracted H2O.
Using a glass column filled with activated silica gel preheated at 120 °C for 24 h, the methylated extracts were fractionated and eluted using n-hexane.
The extracts were separated using the elutropic series of the following four solvent systems including n-hexane, dichloromethane (DCM), and methanol (MeOH). The four systems were: 6 mL of n-hexane, 2 mL of a mixture of n-hexane/dichloromethane (9:1 v/v), 6 mL of dichloromethane, and 5 mL DCM/MeOH (1:1 v/v) to yield an elution rate of 15 mL min1. The DCM/MeOH method allowed the dihydroxy-fatty acids to be eluted. Under a mild steady stream of nitrogen, all fractions were collected and the solvent was removed” instead of “
- Saponification
In the presence of aqueous alkali (KOH), saponification occurs when a fixed oil is converted into soap and alcohol. According to Silva et al. [54], this process was performed before methylation of the fatty acids in each of the four fixed oils. About 10 grams of the hexane fraction were dissolved in about 80 mL of methanol, mixed with about 10 grams of KOH and refluxed for one hour. Then, in a decantation funnel, 240 mL of distilled water was added and the alkaline hydroalcoholic solution was extracted with hexane three times, each for 50 mL As a result of the combination of organic phases, drying over Na2SO4, and concentrating under reduced pressure, the unsaponifiables yielded the pale yellow solid (1.87 g). After acidification with 20% HCl, the hydroalcoholic phase was extracted with acetone three times with 50 mL each. After combining the organic phases, drying over Na2SO4 and concentrating under reduced pressure, 7.15 g of saponifiables were obtained.
- Methylation
The methylation process was done for each of the four fixed oils to enhance the determination of both the fatty acids and the other components [9,54,55] as shown in Figure S3.”
Moreover, the following reference was included in the reference section.
- Hansel, F.A; Bull, I.D.; Evershed, R.P. Gas chromatographic mass spectrometric detection of dihydroxy fatty acids preserved in the ‘bound’ phase of organic residues of archaeological pottery vessels. Rapid Commun. Mass Spectrom. 2011, 25, 1893–1898.
Reviewer comment 17:
Line 395 based on the original numeric system: Which other components? Apparently you have analyse only fatty acids.
Author response 17:
We modified the statement “to enhance the determination of both the fatty acids and the other components” to “to enhance the determination of the fatty acids”
Reviewer comment 18:
Line 464 based on the original numeric system: “For the SG property, there are no statistica differences between the SG values among the four species examined, else the comparisons made within or between species.” I´m sorry but don´t understand this sentence.
Author response 18:
This comment is true. There is a mistake. We corrected this as clear in page 12, line 479 (for new numeric system).
Reviewer comment 19:
Figure 2: a2 is missing and d2 is only “d”
Author response 19:
In Figure 2: a2 was re-wrote and d2 was adjusted.
Further Modifications:
4.2.5. GC-MS-Characterization of Fixed Oils
These fatty acids are represented as methyl or methyl esters due to pretreatments of the fixed oils using the saponification/ methylation route. It has been demonstrated that methylation of oil causes the oil to become more volatile, which subsequently adapts the Programme Temperature Volume of the injector of the GC-MS device [65,66]. There is no difference in fragmentation patterns between methyl esters derivatives of free fatty acids and their parent free fatty acids, except that methyl esters have a higher molecular weight than their parent free fatty acids by 14 grams/moles [66].
Examining Figure 2 concerning to the four fixed oils resulted by the MHPM and EHPM revealed that four fatty acids, namely ricinoleic, oleic, linoleic and palmitic acids were detected, but in different concentrations (Figure 2, S7a-d).
Comparisons of the squeezed fixed oils were performed within and between species to ensure that novel procedures used did not distort their parent properties examined.
Comparisons between species revealed that castor bean had the most abundant ricinoleic acid among the four species studied, while moringa had the highest content of oleic acid. For linoleic acid, it was detected only in castor beans, sunflower and rapeseed, while moringa was found to be free of it. Palmetic acid was detected at all the four species in which castor bean and rapeseed had higher contents of it compared than those for sunflower and moringa. Moreover, comparisons within species indicated that the fixed oils obtained by MHPM gave higher contents of the fatty acids than those produced by EHPM.
Since castor oil is a unique source of ricinoleic acid, a hydroxylated fatty acid [9,47,69], high content of ricinoliec acid can cause the oil to be converted from a non-drying oil to a drying oil by removing the hydroxyl groups in its chain. This is useful in the manufacture of alkyd resin coating [66] and adds to its industrial advantages. The presence of palmitic acid in castor oil may has an economic value to its sodium salt, which is used in the soap and cosmetic industries [52,67].
Effect of microwave on oil extraction
The major advantage of microwave is the reduced time of extraction and energy consumption costs comparing to ordinary techniqyes.
Since the novel MHPM procedure used did not distort parent oil properties examined giving comparatively similar properties, thus it can be justified that using microwaves irradiation is considered as a rapid tool for oil extraction [43].
Enhancing yield and quality of fixed oils by employing microwave irradiation as a heating technique will help industry to more growing and offers mass production due to possibility of worming bigger machines and spaces and fasten production rate [43, 46-48].
The newly introduced references
Seventeen references were added to fill any gap in the manuscript according to the respective reviewers’ comments:
- Farag, R. S.; Hewedi, F.M.; Abu-Raiia, S.H.; Elbaroty, G.S. Comparative study on the deterioration of oils by microwave and conventional heating. Food Prot. 1992, 55: 722-727.
- Takagi, S.; Ienaga, H.; Tsuchiya, C.; Yoshida, H. Microwave roasting effects on the composition of tocopherols and acyl lipids within each structural part and section of a soya bean. Sci. Food Agric. 1999a, 79: 1155-1162.
- Takagi, S.; Yoshida, H.. Microwave heating influences on fatty acid distribution of triacylglycerols and phospholipids in hypocotyls of soybeans (glycine max). Food Chem. 1999b, 66, 345–351.
- Ramanadhan, B. Microwave extraction of essential oils (from black pepper and coriander) at 2.46 Ghz, Canada: University of Saskatchewan, MSc thesis. 2005.
- Uquiche, E.; Jerez, M.; Ortiz, J. Effect of pretreatment with microwaves on mechanical extraction yield and quality of vegetable oil from Chilean hazelnuts (Gevuina avellana Mol). Innov. Food Sci. Emerg. Technol. 2008, 9, 495–500.
- Yoshida, H.; Hirakawa, Y.; Tomiyama, Y.; Mizushina, Y. Effects of microwave treatment on the oxidative stability of peanut (Arachis hypogaea) oils and the molecular species of their triacylglycerols. Eur. Lipid Sci. Technol, 2003, 105, 351-358.
- G.W.; Robertson, J.A. Moisture content/relative humidity equilibrium of high-oil and confectionery type sunflower seed. J. Stored Prod. Res. 1987, 23, 115-118.
- Doan, L.G. Ricin: mechanism of toxicity, clinical manifestations, and vaccine development, A Review.J. Toxicol. 2004, 42, 201– 208.
- Akaranta, O.; Anusiem, A.C.I. A boiresource solvent for extraction of castor oil. Ind Crops Prod. 1996, 5, 273-277.
- Gaul, J.A. Quantitative Calculation of Gas Chromatographic Peaks in Pesticide Residue Analyses. Assoc. Off. Anal. Chem. 1966, 49, 389-399
- Doan, L.G. Ricin: mechanism of toxicity, clinical manifestations, and vaccine development, A Review.J. Toxicol. 2004, 42, 201– 208.
- Akaranta, O.; Anusiem, A.C.I. A boiresource solvent for extraction of castor oil. Ind Crops Prod. 1996, 5, 273-277.
- Hansel, F.A; Bull, I.D.; Evershed, R.P. Gas chromatographic mass spectrometric detection of dihydroxy fatty acids preserved in the ‘bound’ phase of organic residues of archaeological pottery vessels. Rapid Commun. Mass Spectrom. 2011, 25, 1893–1898. Hansel
- Matthaus, B.; Özcan, M.; Al Juhaimi, F. Some rape/canola seed oils: fatty acid composition and tocopherols. Zeitschrift für Naturforschung C. 2016, 71, 73-77. Matthaus et al, 2016
- Chew, S.C. Cold pressed rapeseed (Brassica napus) oil. In Green Technology, Bioactive Compounds, Functionality, and Applications. M. F. Eds., Academic Press. 2020, 65-80. Chew, 2020
- Doan, L.G. Ricin: mechanism of toxicity, clinical manifestations, and vaccine development, A Review.J. Toxicol. 2004, 42, 201– 208. Doan 2004
- Akaranta, O.; Anusiem, A.C.I. A boiresource solvent for extraction of castor oil. Ind Crops Prod. 1996, 5, 273-277. Akaranta, O.; Anusiem 1996.
- Microwave Assisted Drying. 1993. http://ecoursesonline.iasri.res.in/mod/page/view.php?id=882 (accessed on 4 April 2023).
- Rakesh, V.; Seo, Y.; Datta, A. K.; McCarthy, K. L.; McCarthy, M. J. Heat transfer during microwave combination heating: Computational modeling and MRI experiments. Food Nat. Prod. 2010, 56, 2468-2478.
- Watanabe, S.; Karakawa, M.; Hashimoto, O. Temperature of a heated material in a microwave oven considering change of complex relative permittivity. European Microwave Conference (EuMC), Rome, Italy. 2009, 798-801.
The design of materials and methods, results, and discussion sections were enhanced by dividing into the following titles and subtitles:
For the Materials and Methods: The sequence of the main and subtitles are as follow:
- Materials and Methods
2.1. Management plan
2.2. Raw Materials
2.2.1. Preparation of the seeds
2.2.2. Extraction of the Four Fixed Oils
2.2.3. The Hot Pressing Machine (HPM)
2.3. Characterization of the Four Species
2.3.1. Physical Characterization of the Seeds and Oils
2.3.2. Chemical Characterization of the Fixed oils
2.3.2.1. Determination of Iodine Number (IN)
2.3.2.2. Determination of Saponification Value (SV)
2.3.2.3. Determination of Acid Value (AV)
2.3.2.4. Determination of pH Value
2.3.2.5. Determination of the Fatty Acids of the Fixed Oils
- Preparation of the Methyl Esters
- Gas Chromatography-Mass spectrometer (GC–MS)
2.4. Scanning Electron Microscopy (SEM)
2.5. Statistical Design and Analysis
Furthermore, all the equations used to calculate the mean values of the properties of the seeds as well as the fixed oils were collected and presented in a new Table numbered as Table 1 as follow:
Table 1. Calculation of different chemical and physical properties of the seeds and the extracted fixed oils
[7,48,51-53,85].
|
Equation |
Definitions |
|
|
1 MC, % = [(Wads-Wods)/Wods] × 100 2 Scfo, % = (Wmfo+Wrfo)/Wods 3 Ymfo, %= (Wmfo/Wods) ×100 4 Yrfo, % = (Wods-Wrc)/Wods] ×100= (Wrfo / Wods) ×100 5 EL, % = [{Wods-(Wrfo+Wrc)}/Wods]×100 6 Efoe, % = [Wrfo/(Scfo ×Wods)]×100 |
Wads: Weight of air- dried seeds. Wods: Weight of oven-dried seeds, g. Wmfo: Weight of the main fixed oil (g). Wrfo: Weight of recovered fixed oil (from seed’cake), g Wrc = Weight of residual cake after extraction. Scfo: Seed content of fixed oil |
|
|
7 SGfo = (W1-W2)/( W2-W0) |
W0: Weight of an empty bottle. W1: Weight of the bottle filled with fixed oil. W2: Weight of the bottle filled with deionized water. |
|
|
8 IN = 12.69×C × (V1-V2)/W |
C, V1, V2: Parameters of sodium thiosulphate: C: Concentration. V1: The volume used for the blank test. V2: The volume used for the fixed oil. W: The fixed oil weight. |
|
|
9 SV = 56.1N × (V1-V2)/W |
V1: Volume of the solution used for the blank test. V2: Volume of the solution used for fixed oil. N: The actual normality of the HCl used. W: The fixed oil weight. |
|
|
10 AV=5.61(V×N)/W |
V: Volume of KOH, IN mL. N: Normality of KOH. W: Fixed oil weight. |
|
|
11 Yfa = (A1/A2)×100 |
A1: Peak area (mm2) of a certain fatty acid detected at a certain retention time (min.) A2: The overall peak’s areas (mm2) of all fatty acids in the fixed oil. |
|
1 Moisture content of seed, 2 Seed content of fixed oil, 3 Yield of the main fixed oil , 4 Yield of recovered fixed oil, 5 xtraction loss, 6 Efficiency of fixed oil extraction, 7 Specific gravity, 8 Iodine number, 9 Saponification value, 10 Acid value, 11 Yield of fatty acid.
For the results section: The sequence of the main and subtitles are as follow:
- Results
3.1. Physical Properties of seeds
Two paragraphs were included under this title belonging to:
- Moisture content of seeds (MCs), and
- The seed content of fixed oil (Scfo).
3.2. Physical Properties of the Fixed Oils
Without subtitles, six paragraphs were covered this title illustrating each of the following fixed oil properties:
- Yield of the main fixed oil (Ymfo)
- Yield of recovered fixed oil (Yrfo)
- Extraction loss (EL)
- Efficiency of the fixed oil extraction (Efoe)
- Specific gravity of fixed oil (SGfo)
- Refractive Index (RI)
3.3. Chemical Properties of the Fixed Oils
Also, without subtitles, five paragraphs were covered this title illustrating each of the following fixed oil properties:
- Iodine Number (IN)
- Saponification Value (SV)
- Acid Value (AV)
- pH Value
- Yield of fatty acid (Yfa).
3.4. Effect of Microwave Irradiation on Biopolymeric Structured-Tissues (BST): The title was retained without changing.
For the discussion section: The sequence of the main and subtitles are as follow:
- Discussion
4.1. Physical Properties of seeds
Two paragraphs were included under this title belonging to:
- Moisture content of seeds (MCs), and
- The seed content of fixed oil (Scfo).
4.2. Physical Properties of the Fixed Oils
Without subtitles, six paragraphs were covered this title illustrating each of the following fixed oil properties:
- Yield of the main fixed oil (Ymfo)
- Yield of recovered fixed oil (Yrfo)
- Extraction loss (EL)
- Efficiency of the fixed oil extraction (Efoe)
- Specific gravity of fixed oil (SGfo)
- Refractive Index (RI)
4.3. Chemical Properties of the Fixed Oils
Also, without subtitles, five paragraphs were covered this title illustrating each of the following fixed oil properties:
- Iodine Number (IN)
- Saponification Value (SV)
- Acid Value (AV)
- pH Value
- Yield of fatty acid (Yfa).
These old subtitles with their numbers was removed, while their text remained without changing.
4.4. Effect of Microwave Irradiation on Biopolymeric Structured-Tissues (BST): The no. of the title was changed, while its text remained without changing.
4.5. Effect of Ultrastructure of seeds on Fixed Oils Extraction
Also, the no. of the title was changed, while its text remained without changing.

Reviewer 3 Report
The paper <Microwave-assisted extraction of fixed oils from different bi- 2opolymeric structured-tissues> is an interesting one, but the aspects described do not fit the intended purpose. I understood that the main contribution is the modification of one electric-hot pressing machine (EHPM) into a microwave hot pressing machine (MHPM). But, even from the title the things don't match, because the accent is on MW extraction and not on the new invention. So, the first suggestion is to modify the title to highlight the use of a new extraction device.
Major observations:
Why is the relevance of describing the structure of the seeds in the lines 44-51 if the paper is about different extraction methods? I propose to abandon this description!
The same observation for the lines 59-60.
Lines 72-77 fit better after line 160. The authors describe classical methods to extract oil, and then they could introduce some words about microwave assisted extraction.
Lines 171-174 must be reformulated, the authors must describe what they have done and not their expectations.
Line 177: The authors must specify that the management plan is presented in Fig S1 in the supplementary material.
In section 2.2.2 the humidity of the seeds must be reported.
Section 2.2.3: It is difficult to understand the description of the hot pressing machine because the authors refer both to the Figure 1 in the main text and to those in the supplementary material. Please, give a unitary description, or give a brief description in the main text and the whole description in the supplementary material. In Fig. 1 some explanations are missing. What meas <The>….?
2.3.1.1/ It is better to report the yield to the dry seeds, because I presume that the humidity was not the same for all the seeds after drying.
Lines 708-711/ The microwave beam produces an alternate electric field that causes molecules (including water molecules) to rotate so that they align themselves with the alternating electric field produced by the microwave beam. In other words, a wide range of molecules behave like electric dipoles/
Observation:
Due to the fact that water has polar molecules, <especially> instead of <including> must be used in the phrase at line 708. Microwaves (MW) penetrate the molecules proportionally to their dielectric constant (ε) and MW absorption is stronger at high values of the dielectric constant. So, you cannot say that a wide range of molecules behave like electric dipoles, because someone could understand that the MW irradiation is the cause of dipoles existence.
Lines 713-714 <Liquid water is the most efficient medium for microwave heating, while fats and sugars (which have fewer molecular dipole moments) and frozen water (where molecules
are not free to rotate) are less so. >
This sentence is practically identical to the one bellow:
<Microwave heating is most efficient on liquid water, and much less so on fats and sugars (which have less molecular dipole moment), and frozen water (where the molecules are not free to rotate).> http://ecoursesonline.iasri.res.in/mod/page/view.php?id=882/Please reformulate!
Lines 730-732
For the EHPM, heat is transferred mainly by conduction according to the Fourier's
law. On the other hand, for the MHPM, heat is transferred mainly via conduction (accord-
ing to the Fourier's law), convection (according to Newton's law of cooling) and some
amount of heat is transferred via radiation (Kirchhoff's law of thermal radiation).
Observation: The heat transfer mechanism using MW irradiation is more complicated and many research papers describe it. Please, reformulate!
The conclusions are just an enumeration of the aspects already discussed in the text. The authors must highlight their original contribution and the perspectives that their invention can have.
Some minor mistakes:
Line 336 <In this case> must be erased
Lines 338, Why<Mass> with capital letter?
Line 343/ The method was applied only for castor oil?
I suggest to write separately all equations used and to number them.
Standard deviation (SD) is expressed like <mean value ± SD>, why you have used parenthesis?
Line 434/ For the SG property, there are no statistica differences between the SG values among/statistical differences
Line 478/ For the RI characteristic of the fixed oil, no statistica differences were found between/statistical is the correct form
For the RI characteristic of the fixed oil, no statistica differences were found between
Line 690/ Why <For castor oil?> Castor oil is a unique source of …
A microwave hot pressing machine (MHPM) was used to heat the colander of to produce oil from each of castor, sunflower, rapeseed, and moringa seeds and compared to those obtained using ordinary electric-hot pressing machine (EHPM)
|
The SEM study was used to study the surface morphology and anatomical features in biopolymeric structured-tissues [56]. /Please, modify! |
I propose major revision.
Author Response
The Respective Reviewer 3
Notice: In this report, I mean that new numerical system belongs to my revised manuscript based on the 1st round evaluation made by the reviewers, while virgin numerical belongs to the virgin manuscript.
The 1s- modification
The nomenclature was introduced in the article just beneath the conclusion section as follow:
|
Abbreviation |
Definition |
Abbreviation |
Definition |
|
AC |
Alternate current |
MAEO |
Microwave-assisted extracted oil |
|
ACS |
The American Chemical Society |
MFT |
Maximum final temperature |
|
ADB |
Air-dried membranes |
MGU |
Microwave generator unit |
|
AFM |
Atomic force microscopy |
MHPM |
Microwave hot pressing machine |
|
ASTM |
American Society for Testing and Materials |
NDB |
Nanodehydrated-bioplastic membrane |
|
AV |
Acid value |
NIST |
Nanodehydrated-bioplastic membrane |
|
BST |
Biopolymeric Structured-Tissues |
NPS |
Nanometric particle Size |
|
C |
Castor bean (Ricinus communis L.) |
OY |
Oil yield |
|
CI |
Crystallinity index |
PD |
Pore diameter |
|
CLB |
Compound lipid bodies |
pH |
The acidity or basicity number |
|
CW |
Cell walls |
PS |
Particle size |
|
CY |
Cytoplasm |
PubChem |
An open chemistry database managed by the National Institutes of Health (NHI) |
|
DC |
Direct current |
PVA |
Polyvinyl alcohol |
|
DSC |
Differential scanning calorimetry |
R |
Rapeseed (Brassica napus L. |
|
DTA |
Differential thermal analysis |
RI |
Refractive index |
|
EC |
endosperm cells |
RT |
Residence time |
|
EHPO |
Electric hot pressed oil |
S |
Sunflower (Helianthus annuus L.) |
|
EHPM |
Electric-hot pressing machine |
SD |
Standard deviation |
|
FEG |
Field Emission Gun in the SEM |
SEM |
Scanning electron microscopy |
|
FEI |
Field Electron and Ion US-Company |
SEP |
Self-electrostatic peeling |
|
FOY |
Fixed Oil Yield |
SG |
Specific gravity |
|
FTIR |
Fourier transform infrared spectroscopy |
SLB |
Singular lipid bodies |
|
GA |
Gum Arabic |
SP |
Statistical parameters |
|
GC-MS |
Gas Chromatography-Mass spectrometer |
SR |
Surface roughness |
|
GHz |
Frequency |
SV |
Saponification value |
|
HC |
Heat change in µVs/mg |
SWC |
Sinusoidal wave curve |
|
HPM |
Hot pressing machine |
TGA |
Thermogravimetric analysis |
|
HVT |
High voltage transformer |
TR |
Temperature range |
|
IN |
Iodine number |
VFHF |
Vibrated-free horizontal flow |
|
LSD |
Least significant differencce |
VV |
Void volume |
|
M |
Moringa (Moringa oleifera Lam.) |
XRD |
X-Ray diffraction |
His general comments
1- The background of the introduction including relevant references must be improved.
2- The cited references can be improved: It was improved.
3- The methods must be improved: It was improved according to the reviewer 2,3.
4- The conclusions must be improved: It was improved.
5- The research design must be improved: It was improved according to the reviewers 1, 2, and 3.
6- The results’ presentation must be improved: It was improved.
His comments and our response are as follow: It was improved.
Reviewer comment 1:
The paper <Microwave-assisted extraction of fixed oils from different bi- 2opolymeric structured-tissues> is an interesting one, but the aspects described do not fit the intended purpose. I understood that the main contribution is the modification of one electric-hot pressing machine (EHPM) into a microwave hot pressing machine (MHPM). But, even from the title the things don't match, because the accent is on MW extraction and not on the new invention. So, the first suggestion is to modify the title to highlight the use of a new extraction device.
Author response 1:
The title of the manuscript was converted from “Microwave-assisted extraction of fixed oils from different bi-opolymeric structured-tissues” to “A novel microwave hot pressing machine (MHPM) for produc-tion of fixed oils from different biopolymeric structured-tissues”.
Reviewer comment 2:
Why is the relevance of describing the structure of the seeds in the lines 44-51 based on the original numeric system if the paper is about different extraction methods? I propose to abandon this description!
Author response 2:
The next paragraph (found in page 1 and lines 44-51 based on the original numeric system) was deleted: “The seed consists of three components: embryo, endosperm (sometimes perisperm), and seed-coat. Both endosperm and embryo are the products of double fertilization, whereas the seed-coat develops from the maternal, ovular tissues. The seed habit is a significant advancement in the evolution of higher plants. Seed plants show several evolutionary advantages over spore-producing plants. Fertilization takes place within the protective tissues of the mother plant, and during its development the embryo is nourished by the mother plant. Besides, the embryo is covered by the protective seed-coat and often provided with storage material. These factors made the seed plants so successful that they became the dominant component in the terrestrial environments of the earth [2]”.
Moreover, the referred reference “[2]” in this paragraph was included in the next paragraph at line 46.
Reviewer comment 3:
The same observation for the lines 59-60 based on the original numeric system.
Author response 3:
The next paragraph in page 2, lines 59-62 (based on the original numeric system) was moved to page 21, lines 766-768 according to the new numeric system after modification as follow:
Lipids are generally stored as triacylglycerols in oil bodies.Although there are quantitative differences in the accumulation of storage reserves in seeds, it was not clear whether this would also qualitatively affect the fatty acid profiles of triacylglycerols in these seeds [5].
Reviewer comment 4:
Lines 72-77 based on the original numeric system fit better after line 160. The authors describe classical methods to extract oil, and then they could introduce some words about microwave assisted extraction.
Author response 4:
The paragraph “Microwave beams did not alter the parent oil quality comparing to other ordinary methods [7]. As a result of the present investigation, the major advantage of microwave treatment is the reduced time of extraction and energy consumption costs comparing to ordinary methods. Analysis of oil quality indices that oils from both processes (conventional and microwave assisted-extractions) have comparatively similar properties Accordingly, it was justification of microwaves use as a rapid tool for oil extraction process [7,8]” was moved from page 1 /lines 72-78 to Page 4 lines 162-168 in the original revised pdf version (pages 3, 4 / lines 152-158 in the newly version modified by us). Accordingly numbers of references [7,8] were changed to be 42, 43, and the references from 9 to 43 were decreased by number 2, while all the reference numbers beginning from 44 to the higher still constants.
Reviewer comment 5:
Lines 171-174 based on the original numeric system must be reformulated, the authors must describe what they have done and not their expectations.
Author response 5:
We modified the following two paragraphs to be as follow: “Microwave beams did not alter the parent oil quality comparing to other ordinary methods [42]. As a result of the present investigation, the major advantage of micro-wave treatment is the reduced time of extraction and energy consumption costs com-paring to ordinary methods. Analysis of oil quality indices that oils from both process-es (conventional and microwave assisted-extractions) have comparatively similar properties. Accordingly, it was justification of microwaves use as a rapid tool for oil extraction process [42,43].
This study was initiated to evaluate the yield and properties of each of the four fixed oils squeezed using two machines differed in their heating tools (EHPM and MHPM) to ensure that the novel microwave procedure used did not distort the parent quality of fixed oils”.
Both paragraphs are situated in page 4 and lines 152-162.
Reviewer comment 6:
Line 177 based on the original numeric system: The authors must specify that the management plan is presented in Fig S1 in the supplementary material.
Author response 6:
The management plan presented in Fig S1 was specified to its presence in the supplementary material at page 4 in line 165 based on the new numeric system.
Reviewer comment 7:
In section 2.2.2 the humidity of the seeds must be reported.
Author response 7:
The title “2.3.1.1. Moisture Content (MC) of the Seeds” was introduced in page 8, line 290 (new numeric system). Also, the title of “3.1.1. Moisture content of seeds (MC)” was added to line page 12, line 462 (new numeric system). Moreover, For the Discussion section, a paragrapgh concerning to the MC (page 19, lines 639-642) according to the new numeric system.
Reviewer comment 8:
Section 2.2.3: It is difficult to understand the description of the hot pressing machine because the authors refer both to the Figure 1 in the main text and to those in the supplementary material. Please, give a unitary description, or give a brief description in the main text and the whole description in the supplementary material. In Fig. 1 some explanations are missing. What means <The>….?
Author response 8:
For the word “The” refer to “The guideline” but the rectangle containing the words was stretched upon its paste, since all the figure are still JPG not PDF. However, this mistake was corrected.
For the other comment in the same point, we corrected this section based on this intelligent comment. We moved supplementary-figures (S4, S5, S6) to be included in the main text of the manuscript giving a unitary description hot pressing machine.
Section 2.2.3: It is difficult to understand the description of the hot pressing machine because the authors refer both to the Figure 1 in the main text and to those in the supplementary material. Please, give a unitary description, or give a brief description in the main text and the whole description in the supplementary material.
Reviewer comment 9:
2.3.1.1/ It is better to report the yield to the dry seeds, because I presume that the humidity was not the same for all the seeds after drying.
Author response 9:
In the Materials and Methods section:
We added the next title under the section 2.3.1 and took the subtitle of 2.3.1.1. Moisture Content (MC) of the Seeds (Page 8, starting at line 291-296, the new number of the newly revised and modified version)”:
About 100g of cleaned intact seeds was dried in an oven at 90°C, weighed at 1 hour intervals until obtaining a constant weight after about 6 hours. The moisture content was calculated using the following formula [41]:
MC, % = [(W1-W2)/W2] × 100
Where W1 and W2 are the air-dried and oven-dried weights of sample, respectively.
In the results section,
We added the next title under the section 2.3.1 as well as the following paragraph “The MC of castor beans were found to be 3.6% [41,61,62] as adapted to that determined by Muzenda et al [42] and differed than those reported in the literature findings of 4.15% [60] and 5-7% [91] (page 11, lines 460-).
Moreover, a modification was done for Table 1 by adding two new rows as the 1st and 2nd upper rows. The original title of Table 1 present at (page 11 and lines 444-446) according to the original revised version sent from the editor” Table 1. Mean values1,2 of physical and chemical properties of the four fixed oils, namely castor (C), sunflower (S), rapeseed (R), and moringa (M) extracted by microwave-hot pressing machine (MHPM) and electric hot pressing machine as compared to the ASTM specifications.” found in page 11 and lines 451-454 of the new sequence was changed to be “Table 1. Mean values1,2 of moisture content (MC) of each of air-dried (AD) and oven-dried (OD) seeds, physical and chemical properties of the four fixed oils, namely castor (C), sunflower (S), rapeseed (R), and moringa (M) extracted by microwave-hot pressing machine (MHPM) and electric hot pressing machine as compared to the ASTM specifications”.
In the discussion section
We added the next title under the section 4.1.1 (page 19, line 451-459).
The variation could be attributed to the difference in the nature of beans from different locations [41] and/or seed macro-structure including hull to kernel weight ratio, hull thickness, and their oil content [62].
Moreover, two relevant references were introduced as follow
- Doan, L.G. Ricin: mechanism of toxicity, clinical manifestations, and vaccine development, A Review.J. Toxicol. 2004, 42, 201– 208.
- Akaranta, O.; Anusiem, A.C.I. A boiresource solvent for extraction of castor oil. Ind Crops Prod. 1996, 5, 273-277.
- G.W.; Robertson, J.A. Moisture content/relative humidity equilibrium of high-oil and confectionery type sunflower seed. J. Stored Prod. Res., 23, 115-118.
- Microwave Assisted Drying. 1993. http://ecoursesonline.iasri.res.in/mod/page/view.php?id=882 (accessed on 4 April 2023).
Reviewer comment 10:
Lines 708-711 based on the original numeric system / The microwave beam produces an alternate electric field that causes molecules (including water molecules) to rotate so that they align themselves with the alternating electric field produced by the microwave beam. In other words, a wide range of molecules behave like electric dipoles/ Due to the fact that water has polar molecules, <especially> instead of <including> must be used in the phrase at line 708 based on the original numeric system. Microwaves (MW) penetrate the molecules proportionally to their dielectric constant (ε) and MW absorption is stronger at high values of the dielectric constant. So, you cannot say that a wide range of molecules behave like electric dipoles, because someone could understand that the MW irradiation is the cause of dipoles existence.
Author response 10:
We changed the word “including” into “especially” in the line 709, 9age 21 (virgin numerical system) equivalent to the new numerical system (line 730, page 22).
Reviewer comment 11:
Lines 713-714 based on the original numeric system <Liquid water is the most efficient medium for microwave heating, while fats and sugars (which have fewer molecular dipole moments) and frozen water (where molecules are not free to rotate) are less so. > This sentence is practically identical to the one bellow:
<Microwave heating is most efficient on liquid water, and much less so on fats and sugars (which have less molecular dipole moment), and frozen water (where the molecules are not free to rotate).> http://ecoursesonline.iasri.res.in/mod/page/view.php?id=882/Please reformulate!
Author response 11:
The commended two paragraphs were confounded together and the resultant paragraph was reformulated to be (as shown in page 22, lines 727-734, new numerical system):
In microwave heating, microwaves penetrate to the interior of biopolymeric structured-tissues including but not limited to seeds as well as foods affecting their dipolar molecules of water, fats, or sugars. Accordingly, heat is generated because of dipolar molecules absorb electromagnetic radiation and, leading to their intensive vibration producing friction that leads to rapid temperature rising and consequently efficient water evaporation and/or melting fats. This results in a greatly increased vapor pressure differential between the center and surface of the biopolymeric tissue, allowing fast transfer of moisture and or essential oil out of the tissue. Hence, microwave drying as well as microwave hot pressing technique is rapid, more uniform and energy efficient compared to ordinary ones [48,84].
Reviewer comment 12:
Lines 730-732 based on the original numeric system: For the EHPM, heat is transferred mainly by conduction according to the Fourier's law. On the other hand, for the MHPM, heat is transferred mainly via conduction (according to the Fourier's law), convection (according to Newton's law of cooling) and some amount of heat is transferred via radiation (Kirchhoff's law of thermal radiation). Observation: The heat transfer mechanism using MW irradiation is more complicated and many research papers describe it. Please, reformulate!
Author response 12:
Heat transfer
The following paragraph was changed “For the EHPM, heat energy and entropy are transferred mainly by conduction according to the Fourier's law. On the other hand, for the MHPM, heat is transferred mainly via conduction (according to the Fourier's law), convection (according to Newton's law of cooling) and some amount of heat is transferred via radiation (Kirchhoff's law of thermal radiation).
Conduction heat is responsible for moving heat from the surficial regions of the oily seed tissues to their cores, while convection combines conduction heat transfer and circulation to force molecules in the air to move from hottest zones to cooler ones. Moreover, radiation is the process where heat and light waves strike and penetrate your food. As such, there is no direct contact between the heat source and the cooking food.
Accordingly, using the microwave hot pressing machine exhibits an ideal tool for heat transfer and subsequently heat the oily seed tissues and increases the fixed oil yield [84-86].
Three references were added to help us for checking and improving this paragraph scientifically:
- Anonymous. Microwave Assisted Drying. 1993. http://ecoursesonline.iasri.res.in/mod/page/view.php?id=882 (accessed on 4 April 2023).
- Rakesh, V.; Seo, Y.; Datta, A. K.; McCarthy, K. L.; McCarthy, M. J. Heat transfer during microwave combination heating: Computational modeling and MRI experiments. Bioeng. Food Nat. Prod. 2010, 56, 2468-2478.
- Watanabe, S.; Karakawa, M.; Hashimoto, O. Temperature of a heated material in a microwave oven considering change of complex relative permittivity. European Microwave Conference (EuMC), Rome, Italy. 2009, 798-801.
Reviewer comment 13:
The conclusions are just an enumeration of the aspects already discussed in the text. The authors must highlight their original contribution and the perspectives that their invention can have.
Author response 13:
In conclusion, the point no. 6 was deleted, and the followed points were re-numbered in a correct sequence.
Reviewer comment 14:
Line 336 <In this case> must be erased
Author response 14:
The phrase” in this case” in the beginning of line 336, page 8 (old numerical system) was deleted and we added “where” instead of it (line 354, page 11 in new numerical system).
Reviewer comment 15:
Lines 338 based on the original numeric system, Why<Mass> with capital letter?
Author response 15:
The word “Mass” was corrected to be” mass”.
Reviewer comment 16:
Line 343 based on the original numeric system / The method was applied only for castor oil?
Author response 16:
The following statement ”About 2 g of a fixed oil was added to” was introduced (line 361, page 11, new numerical system) instead of “About 0.5 g of castor oil dissolved in ethanol was mixed with” ((line 343, page 9, virgin numerical system).
Reviewer comment 17:
I suggest to write separately all equations used and to number them.
Author response 17:
It is very interested idea. Accordingly, I collect all equations into Figure 1 presented at the Materials and Methods section found in page 10, lines 300-305.
Reviewer comment 18:
Standard deviation (SD) is expressed like <mean value ± SD>, why you have used parenthesis?
Author response 18:
The parenthesis were deleted in Table 1 and the sign ± was added to each mean value.
Reviewer comment 19:
Line 434 based on the original numeric system / For the SG property, there are no statistica differences between the SG values among/statistical differences.
Author response 19:
statistica was corrected to be “ statistical” in line 470 (the new sequence).
Reviewer comment 20:
Line 478 based on the original numeric system / For the RI characteristic of the fixed oil, no statistica differences were found between/statistical is the correct form
Author response 20:
statistica was corrected to be “ statistical” in line 484 (the new sequence).
Reviewer comment 21:
Line 690 based on the original numeric system / Why <For castor oil?> Castor oil is a unique source of …
Author response 21:
line 696-697 based on the original numeric system was changed into: I. Castor oil is a unique source of ricinoleic acid, a hydroxylated fatty acid as clear at Figures 2a1,a2 [9,47,66].
Reviewer comment 22:
A microwave hot pressing machine (MHPM) was used to heat the colander of to produce oil from each of castor, sunflower, rapeseed, and moringa seeds and compared to those obtained using ordinary electric-hot pressing machine (EHPM).
Author response 22:
In line 15, page 1, the statement ”…….(MHPM) was used to heat the colander of to produce from each of castor” was corrected to be “MHPM) was used to heat the colander to produce fixed oil from each of castor….”
Reviewer comment 23:
|
The SEM study was used to study the surface morphology and anatomical features in biopolymeric structured-tissues [56]. /Please, modify! |
Author response 23:
“The SEM study was used to study the surface morphology” was changed to be “The SEM imaging was used to study the surface morphology”
Further Modifications:
4.2.5. GC-MS-Characterization of Fixed Oils
These fatty acids are represented as methyl or methyl esters due to pretreatments of the fixed oils using the saponification/ methylation route. It has been demonstrated that methylation of oil causes the oil to become more volatile, which subsequently adapts the Programme Temperature Volume of the injector of the GC-MS device [65,66]. There is no difference in fragmentation patterns between methyl esters derivatives of free fatty acids and their parent free fatty acids, except that methyl esters have a higher molecular weight than their parent free fatty acids by 14 grams/moles [66].
Examining Figure 2 concerning to the four fixed oils resulted by the MHPM and EHPM revealed that four fatty acids, namely ricinoleic, oleic, linoleic and palmitic acids were detected, but in different concentrations (Figure 2, S7a-d).
Comparisons of the squeezed fixed oils were performed within and between species to ensure that novel procedures used did not distort their parent properties examined.
Comparisons between species revealed that castor bean had the most abundant ricinoleic acid among the four species studied, while moringa had the highest content of oleic acid. For linoleic acid, it was detected only in castor beans, sunflower and rapeseed, while moringa was found to be free of it. Palmetic acid was detected at all the four species in which castor bean and rapeseed had higher contents of it compared than those for sunflower and moringa. Moreover, comparisons within species indicated that the fixed oils obtained by MHPM gave higher contents of the fatty acids than those produced by EHPM.
Since castor oil is a unique source of ricinoleic acid, a hydroxylated fatty acid [9,47,69], high content of ricinoliec acid can cause the oil to be converted from a non-drying oil to a drying oil by removing the hydroxyl groups in its chain. This is useful in the manufacture of alkyd resin coating [66] and adds to its industrial advantages. The presence of palmitic acid in castor oil may has an economic value to its sodium salt, which is used in the soap and cosmetic industries [52,67].
Effect of microwave on oil extraction
The major advantage of microwave is the reduced time of extraction and energy consumption costs comparing to ordinary techniqyes.
Since the novel MHPM procedure used did not distort parent oil properties examined giving comparatively similar properties, thus it can be justified that using microwaves irradiation is considered as a rapid tool for oil extraction [43].
Enhancing yield and quality of fixed oils by employing microwave irradiation as a heating technique will help industry to more growing and offers mass production due to possibility of worming bigger machines and spaces and fasten production rate [43, 46-48].
The newly introduced references
Seventeen references were added to fill any gap in the manuscript according to the respective reviewers’ comments:
- Farag, R. S.; Hewedi, F.M.; Abu-Raiia, S.H.; Elbaroty, G.S. Comparative study on the deterioration of oils by microwave and conventional heating. Food Prot. 1992, 55: 722-727.
- Takagi, S.; Ienaga, H.; Tsuchiya, C.; Yoshida, H. Microwave roasting effects on the composition of tocopherols and acyl lipids within each structural part and section of a soya bean. Sci. Food Agric. 1999a, 79: 1155-1162.
- Takagi, S.; Yoshida, H.. Microwave heating influences on fatty acid distribution of triacylglycerols and phospholipids in hypocotyls of soybeans (glycine max). Food Chem. 1999b, 66, 345–351.
- Ramanadhan, B. Microwave extraction of essential oils (from black pepper and coriander) at 2.46 Ghz, Canada: University of Saskatchewan, MSc thesis. 2005.
- Uquiche, E.; Jerez, M.; Ortiz, J. Effect of pretreatment with microwaves on mechanical extraction yield and quality of vegetable oil from Chilean hazelnuts (Gevuina avellana Mol). Innov. Food Sci. Emerg. Technol. 2008, 9, 495–500.
- Yoshida, H.; Hirakawa, Y.; Tomiyama, Y.; Mizushina, Y. Effects of microwave treatment on the oxidative stability of peanut (Arachis hypogaea) oils and the molecular species of their triacylglycerols. Eur. Lipid Sci. Technol, 2003, 105, 351-358.
- G.W.; Robertson, J.A. Moisture content/relative humidity equilibrium of high-oil and confectionery type sunflower seed. J. Stored Prod. Res. 1987, 23, 115-118.
- Doan, L.G. Ricin: mechanism of toxicity, clinical manifestations, and vaccine development, A Review.J. Toxicol. 2004, 42, 201– 208.
- Akaranta, O.; Anusiem, A.C.I. A boiresource solvent for extraction of castor oil. Ind Crops Prod. 1996, 5, 273-277.
- Gaul, J.A. Quantitative Calculation of Gas Chromatographic Peaks in Pesticide Residue Analyses. Assoc. Off. Anal. Chem. 1966, 49, 389-399
- Doan, L.G. Ricin: mechanism of toxicity, clinical manifestations, and vaccine development, A Review.J. Toxicol. 2004, 42, 201– 208.
- Akaranta, O.; Anusiem, A.C.I. A boiresource solvent for extraction of castor oil. Ind Crops Prod. 1996, 5, 273-277.
- Hansel, F.A; Bull, I.D.; Evershed, R.P. Gas chromatographic mass spectrometric detection of dihydroxy fatty acids preserved in the ‘bound’ phase of organic residues of archaeological pottery vessels. Rapid Commun. Mass Spectrom. 2011, 25, 1893–1898. Hansel
- Matthaus, B.; Özcan, M.; Al Juhaimi, F. Some rape/canola seed oils: fatty acid composition and tocopherols. Zeitschrift für Naturforschung C. 2016, 71, 73-77. Matthaus et al, 2016
- Chew, S.C. Cold pressed rapeseed (Brassica napus) oil. In Green Technology, Bioactive Compounds, Functionality, and Applications. M. F. Eds., Academic Press. 2020, 65-80. Chew, 2020
- Doan, L.G. Ricin: mechanism of toxicity, clinical manifestations, and vaccine development, A Review.J. Toxicol. 2004, 42, 201– 208. Doan 2004
- Akaranta, O.; Anusiem, A.C.I. A boiresource solvent for extraction of castor oil. Ind Crops Prod. 1996, 5, 273-277. Akaranta, O.; Anusiem 1996.
- Microwave Assisted Drying. 1993. http://ecoursesonline.iasri.res.in/mod/page/view.php?id=882 (accessed on 4 April 2023).
- Rakesh, V.; Seo, Y.; Datta, A. K.; McCarthy, K. L.; McCarthy, M. J. Heat transfer during microwave combination heating: Computational modeling and MRI experiments. Food Nat. Prod. 2010, 56, 2468-2478.
- Watanabe, S.; Karakawa, M.; Hashimoto, O. Temperature of a heated material in a microwave oven considering change of complex relative permittivity. European Microwave Conference (EuMC), Rome, Italy. 2009, 798-801.
The design of materials and methods, results, and discussion sections were enhanced by dividing into the following titles and subtitles:
For the Materials and Methods: The sequence of the main and subtitles are as follow:
- Materials and Methods
2.1. Management plan
2.2. Raw Materials
2.2.1. Preparation of the seeds
2.2.2. Extraction of the Four Fixed Oils
2.2.3. The Hot Pressing Machine (HPM)
2.3. Characterization of the Four Species
2.3.1. Physical Characterization of the Seeds and Oils
2.3.2. Chemical Characterization of the Fixed oils
2.3.2.1. Determination of Iodine Number (IN)
2.3.2.2. Determination of Saponification Value (SV)
2.3.2.3. Determination of Acid Value (AV)
2.3.2.4. Determination of pH Value
2.3.2.5. Determination of the Fatty Acids of the Fixed Oils
- Preparation of the Methyl Esters
- Gas Chromatography-Mass spectrometer (GC–MS)
2.4. Scanning Electron Microscopy (SEM)
2.5. Statistical Design and Analysis
Furthermore, all the equations used to calculate the mean values of the properties of the seeds as well as the fixed oils were collected and presented in a new Table numbered as Table 1 as follow:
Table 1. Calculation of different chemical and physical properties of the seeds and the extracted fixed oils
[7,48,51-53,85].
|
Equation |
Definitions |
|
|
1 MC, % = [(Wads-Wods)/Wods] × 100 2 Scfo, % = (Wmfo+Wrfo)/Wods 3 Ymfo, %= (Wmfo/Wods) ×100 4 Yrfo, % = (Wods-Wrc)/Wods] ×100= (Wrfo / Wods) ×100 5 EL, % = [{Wods-(Wrfo+Wrc)}/Wods]×100 6 Efoe, % = [Wrfo/(Scfo ×Wods)]×100 |
Wads: Weight of air- dried seeds. Wods: Weight of oven-dried seeds, g. Wmfo: Weight of the main fixed oil (g). Wrfo: Weight of recovered fixed oil (from seed’cake), g Wrc = Weight of residual cake after extraction. Scfo: Seed content of fixed oil |
|
|
7 SGfo = (W1-W2)/( W2-W0) |
W0: Weight of an empty bottle. W1: Weight of the bottle filled with fixed oil. W2: Weight of the bottle filled with deionized water. |
|
|
8 IN = 12.69×C × (V1-V2)/W |
C, V1, V2: Parameters of sodium thiosulphate: C: Concentration. V1: The volume used for the blank test. V2: The volume used for the fixed oil. W: The fixed oil weight. |
|
|
9 SV = 56.1N × (V1-V2)/W |
V1: Volume of the solution used for the blank test. V2: Volume of the solution used for fixed oil. N: The actual normality of the HCl used. W: The fixed oil weight. |
|
|
10 AV=5.61(V×N)/W |
V: Volume of KOH, IN mL. N: Normality of KOH. W: Fixed oil weight. |
|
|
11 Yfa = (A1/A2)×100 |
A1: Peak area (mm2) of a certain fatty acid detected at a certain retention time (min.) A2: The overall peak’s areas (mm2) of all fatty acids in the fixed oil. |
|
1 Moisture content of seed, 2 Seed content of fixed oil, 3 Yield of the main fixed oil , 4 Yield of recovered fixed oil, 5 xtraction loss, 6 Efficiency of fixed oil extraction, 7 Specific gravity, 8 Iodine number, 9 Saponification value, 10 Acid value, 11 Yield of fatty acid.
For the results section: The sequence of the main and subtitles are as follow:
- Results
3.1. Physical Properties of seeds
Two paragraphs were included under this title belonging to:
- Moisture content of seeds (MCs), and
- The seed content of fixed oil (Scfo).
3.2. Physical Properties of the Fixed Oils
Without subtitles, six paragraphs were covered this title illustrating each of the following fixed oil properties:
- Yield of the main fixed oil (Ymfo)
- Yield of recovered fixed oil (Yrfo)
- Extraction loss (EL)
- Efficiency of the fixed oil extraction (Efoe)
- Specific gravity of fixed oil (SGfo)
- Refractive Index (RI)
3.3. Chemical Properties of the Fixed Oils
Also, without subtitles, five paragraphs were covered this title illustrating each of the following fixed oil properties:
- Iodine Number (IN)
- Saponification Value (SV)
- Acid Value (AV)
- pH Value
- Yield of fatty acid (Yfa).
3.4. Effect of Microwave Irradiation on Biopolymeric Structured-Tissues (BST): The title was retained without changing.
For the discussion section: The sequence of the main and subtitles are as follow:
- Discussion
4.1. Physical Properties of seeds
Two paragraphs were included under this title belonging to:
- Moisture content of seeds (MCs), and
- The seed content of fixed oil (Scfo).
4.2. Physical Properties of the Fixed Oils
Without subtitles, six paragraphs were covered this title illustrating each of the following fixed oil properties:
- Yield of the main fixed oil (Ymfo)
- Yield of recovered fixed oil (Yrfo)
- Extraction loss (EL)
- Efficiency of the fixed oil extraction (Efoe)
- Specific gravity of fixed oil (SGfo)
- Refractive Index (RI)
4.3. Chemical Properties of the Fixed Oils
Also, without subtitles, five paragraphs were covered this title illustrating each of the following fixed oil properties:
- Iodine Number (IN)
- Saponification Value (SV)
- Acid Value (AV)
- pH Value
- Yield of fatty acid (Yfa).
These old subtitles with their numbers was removed, while their text remained without changing.
4.4. Effect of Microwave Irradiation on Biopolymeric Structured-Tissues (BST): The no. of the title was changed, while its text remained without changing.
4.5. Effect of Ultrastructure of seeds on Fixed Oils Extraction
Also, the no. of the title was changed, while its text remained without changing.

Round 2
Reviewer 1 Report
The authors have made good efforts to improve the clarity and impact of the manuscript, according to the reviewers' comments and suggestions.
Author Response
We thanks the respective reviewer 1 for his recognizing our efforts. For the research design we have already changed this design as a revolution based on his valuable comments in the 1st round stage. I am ready to make any changes that can enhance the article level.

Reviewer 2 Report
The English language has been clearly improved, as well as the general structure of the article, however there are still some details that need attention.
Line 154-160: This paragraph could be better located in the conclusions section not in the introduction.
Line 212: Probably you meant “warming” instead of “worming”.
Line 217: The same for “warm” instead “worm”
Line 306-307: Please, explain this sentence “If a fixed oil has an unknown physical parameter, it can be determined and compared with the cited standard”.
Line 329-330: Please explain the sentence “The fine particle size of the seeds is the first phase in cell distraction.”
Line 334: Is there anything missing after “The recovered fixed”?
Line 339 to 344: Please revise the whole paragraph.
Line 473-475: Please explain the sentence “The comparisons between the mean values of these properties were restricted to those within species not between species. Accordingly the comparisons were done only between the fixed oils extracted using the microwave-hot pressing machine (MHPM) and the electric hot pressing machine (EHPM)”. The text compares both the differences between oils of different species and oils obtained by different methods.
Line 663: Please delete “Showing”
Line 751: Please explain the sentence “Several researcher have been trying to use [48,78-83]”
Please revise the figure numbers, when you talk about figure 4 in the text it seems that you are referring to figure 3 and when you talk about figure 5 it seems that you are referring to figure 4. I couldn´t find figure 5.
Author Response
Comment 1
Line 154-160: This paragraph could be better located in the conclusions section not in the introduction section.
Response 1
This paragraph was deleted from the introduction (Line 154-160, page 4) due to has repeated findings found originally in the conclusion section.
Comment 2
Line 212: Probably you meant “warming” instead of “worming”.
Response 2
Yes, I meant warming not worming. I corrected this mistake (line 205, page 5 based on the new numerical system).
Comment 3
Line 217: The same for “warm” instead “worm”
Response 3
Ok, I corrected worm to be warm in line 212 (origin numerical system) that is equivalent to line 210 (the new numerical system)
Comment 4
Line 306-307: Please, explain this sentence “If a fixed oil has an unknown physical parameter, it can be determined and compared with the cited standard”.
Response 4
This is a mistake, accordingly, the referred statement was deleted and was substituted by the following one”
“These properties were determined and compared with the cited standard [7,51].”
This is found in page 9, lines 299-300 (the new numerical system).
Comment 5
Line 329-330 (old numeric.): Please explain the sentence “The fine particle size of the seeds is the first phase in cell distraction.”
Response 5
This range of the seeds’ particle size facilitates the diffusion of the soluble compounds and the release of the oil [52,53] as presented in page 10, lines 321-322, new numeric.
Comment 6
Line 334: Is there anything missing after “The recovered fixed”?
Response 5
Yes, the statement is incomplete, it was corrected as follow:
“The recovered fixed oil was calculated in the manner indicated in Table 1”.
Comment 7
Line 339 to 344: Please revise the whole paragraph.
“Line 473-475: Please explain the sentence “The comparisons between the mean values of these properties were restricted to those within species not between species. Accordingly the comparisons were done only between the fixed oils extracted using the microwave-hot pressing machine (MHPM) and the electric hot pressing machine (EHPM)”. The text compares both the differences between oils of different species and oils obtained by different methods “.
Response 7
As clear in page 13, lines 466-471, the following paragraph was modified”
“The comparisons between the mean values of these properties were extended to those within species as well as between species. Accordingly the comparisons were done between the fixed oils extracted using the microwave-hot pressing machine (MHPM) and the electric hot pressing machine (EHPM) as well as between the oily species at the same level of machinery used.”
Comment 8
Line 663: Please delete “Showing”
Response 8
The word “showing” in line 664 (page 19) was deleted (new numeric.).
Comment 9
Line 751: Please explain the sentence “Several researcher have been trying to use [48,78-83]”
Response 9
The referred statement is not complete, so we corrected it as follow:
“Several researcher have been trying to use microwave irradiation a sustainable non-contact heating source for oil extraction from aromatic and oily plants[48,78-83].” as shown in page 21, lines 744-745 (new numeric.)
Comment 10
Please revise the figure numbers, when you talk about figure 4 in the text it seems that you are referring to figure 3 and when you talk about figure 5 it seems that you are referring to figure 4. I couldn´t find figure 5.
Response 10
The following paragraph was removed due to its equation is presented aleady in Table 1
- “: IN = 12.69 × C(V_1-V_2)/m, where C represents the concentration of sodium thiosulphate used, V1 indicates the volume of sodium thiosulphate used for the blank, V2 indicates the volume of sodium thiosulphate used for the oil, and m represents the mass of the fixed oil” in page 11, lines 636-366 (new numeric. system).
- In Table 1, the upper 3 rd statement under the definition title was converted from italic into normal letters as highlighted by blue color.
- In the title of Figure (page 5, lines 216-219):
The paragraph “Figure 1. The microwave-hot pressing machine (MHPM) used for extraction of the four fixed oils: a) An overall front image for the HPM before connecting the microwave source, b) An overall front image for the MHPM, and c) The waveguide that used for directing microwave beam to the extrusion head. The microwave generator unit (MGU) used for MHPM” was corrected to be “Figure 1. The microwave-hot pressing machine (MHPM) used for extraction of the four fixed oils: a) An overall front image for the HPM before connecting the microwave source, b) An overall front image for the MHPM showing the colander, the waveguide that used for directing microwave beam to the extrusion head, and the microwave generator unit (MGU).”
- “Figures 1a,b, 2a,c, 3a,c, 4c” was corrected to be “Figures 1a,c, 2a,c, 3a,c, 4e” (page 6, line 229)
- “Figure 4c,d” was corrected to be “4c” (page 8, line 274).
- The statement “Moreover, their chromatograms are presented at Figure S4. The percentage allocations of the fatty acids in the fixed oils are presented at Figure 5.” was added in page 12, lines 440-442.
- Pages 15, 16, 17: the word Figure 2 was corrected to be Figure 5 as highlighted in blue.
- Page 18: “Figure 7” was corrected to “Figure S4”.
- Page 18: line 644, “Figure 4” was corrected to “Figure 6”.
- Page 18: line 648, “Figure 3” was corrected to “Figure 6”.
- Page 19: line 658, “Figure 4” was corrected to “Figure 7”.
- Page 19: line 662, “Figure 4” was corrected to “Figure 7”.
In Supplementary material:
In Figure S3, for the upper 2nd rectangle, its content was adjusted to size 10 instead of 12 as well as for the lowest rectangle, its contents of words were converted from bold to normal letters.
In the title of Figure S4, “of:a)” was adjusted to be “of: a).

Round 3
Reviewer 2 Report
Line 671: Please revise the heading of section 4.1